# Effective, Stable and Efficient Unsupervised Image Outlier Detection via Distance Ensemble Learning

## Abstract

To automatically and efficiently identify whether visual systems involve outliers (anomalies) is an important research topic. Although there has been rapid progress in the efficacy of unsupervised image outlier detection, the instability and complexity of the state-of-the-art (SOTA) methods is still a notable challenge. In this work, we explain the instability problem derived from the mainstream single method-fits-multiple scenarios paradigm, which results in performance fluctuations across different target dataset domains and varying outlier ratios. Therefore, ensembling multiple methods seems necessary. Nevertheless, traditional ensemble learners such as stacking and boosting are less effective without any supervision and are often time-consuming. Such that, we introduce a novel and lightweight distance ensemble learning (DEL) framework featuring self-selection strategies over a series of distance-based methods. Specifically, by exploring a specific property of the high-dimensional space, we propose the normalized Euclidean distance relative to the mean of the target dataset as a reliable baseline. Building upon this baseline method, we enhance it with a conditional bilateral distance metric to achieve stability across diverse dataset domains at low outlier ratios. Furthermore, to address the mean-shift problem encountered by the advanced baseline at high outlier ratios, we integrate it with a high-ratio specific distance transformer, called Shell-Re. This subsequent integration effectively mitigates the advanced baseline's instability across a wide range of outlier ratios. Overall, our approach achieves SOTA results on various challenging benchmarks while offering inference speeds that are orders of magnitude faster.

## 1 Introduction

Unsupervised[1] image outlier detection is about identifying those images that deviate from the predominant mechanism within an unlabelled and contaminated target dataset. Currently, the mainstream approaches follow a single method-fits-multiple scenarios paradigm, i.e., learning an outlier score function to estimate the likelihood of an instance belonging to outliers. Despite the rapid progress, both the high structure complexity of SOTA methods (Lai et al., 2019; Lin et al., 2021; Li et al., 2022; Wang et al., 2023) and their efficacy instability, i.e., detection accuracy fluctuations on various real-world scenarios remain significant challenges.

For a clear discussion, we decompose instability into two parts: domain instability and outlier (contamination) ratio (Perini et al., 2023) instability. First, we contend that unsupervised image outlier detection inherently encompasses two distinct sub-tasks depending on the outlier ratio. Specifically, when the ratio is low, the primary focus of the target dataset is on inliers, which motivates manifold-based methods (Zhou & Paffenroth, 2017; Zong et al., 2018; Lai et al., 2019; Lin et al., 2022) to identify inliers' distributions. However, as the outlier ratio increases, these inlier manifolds undergo significant shifts, leading to a decrease in detection accuracy. In such scenarios, techniques that facilitate self-outlier exposures, such as Shell-Re (Lin et al., 2021) become more advantageous due to the presence of a large number of outliers. The contrast is shown in Fig. 1. Furthermore, empirical studies (Han et al., 2022) validate that instability also arises from dataset domains.

Based on the above facts, a natural question arises: Is it possible that an efficient method can achieve the overall SOTA performance across all scenarios? To this end, this work introduces a novel ensemble learning

---

[1] Unsupervised in this area refers to the training data is unavailable.

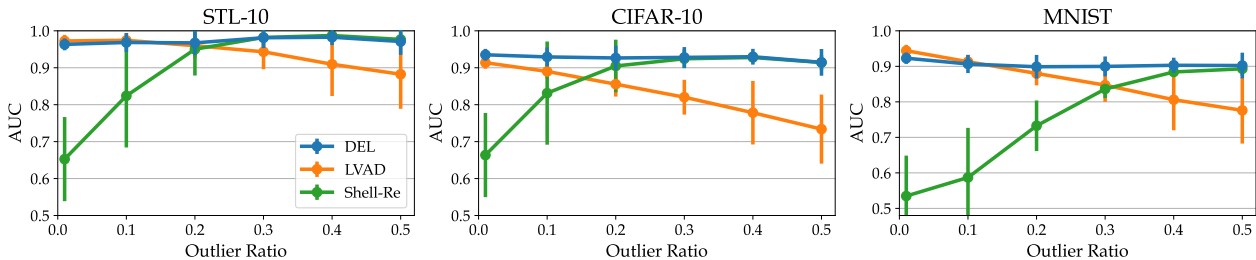

Figure 1: Average AUC results across a wide range of outlier ratios on three datasets. Our proposed DEL demonstrates remarkable efficacy and stability compared to Shell-Re (TPAMI-21) and LVAD (ECCV-22).

framework: Distance Ensemble Learning (DEL) with self-selection on a series of distance-based methods. Compared with classical ensemble learning methods such as stacking and boosting which are time-consuming and less effective without any supervision, DEL derives from an interesting intuition of the high-dimensional visual space, i.e., the normalized Euclidean distance transform relative to the target dataset's mean will be a natural discriminative outlier score function. Despite this distance transform still suffers from the inevitable instability over various dataset domains and outlier ratios, as it is simple and explainable, we take it as our baseline and subsequently improve its stability with a two-stage ensemble learning (ensembling) strategy.

Specifically, we first introduce a conditional bilateral distance metric and integrate it with the baseline, which significantly improves the stability of various benchmark dataset domains. If the outlier ratio is low, the advanced baseline exhibits promising detection accuracy while its efficacy undergoes a decrease when the outlier ratio becomes higher, termed the mean-shift problem. To tackle this issue, we subsequently ensemble the advanced baseline with Shell-Re, a high-ratio specific distance-based outlier detector. Since Shell-Re is merely effective on high outlier ratio scenarios and satisfies the structure consistency with our advanced baseline method, the ranking-index similarity between the advanced baseline and Shell-Re will be positively correlated with the outlier ratio. So it is flexible to ensemble the advanced baseline and Shell-Re. Overall, our proposed DEL framework will be stable over various dataset domains and varying outlier ratios.

To comprehensively assess the performance of our DEL framework, we subject it to rigorous testing across nine benchmark datasets and three feature representations, while spanning six outlier ratios. With its remarkable simplicity, our experimental results demonstrate it achieves the overall SOTA results of ranking accuracy. More importantly, the estimated outlier ratio can be a powerful prior knowledge for ensembling existing deep learning or statistical methods, and open a further perspective that merely centers on designing methods for either low-ratio or high-ratio scenarios. The main contributions of this paper can be summarised as follows:

- First of all, we explore the instability reasons (various target dataset domains and varying outlier ratios) of the unsupervised image outlier detection task and indicate the necessity of ensemble learning.
- Our approach, termed DEL, involves a careful selection and ensembling of distance-based methods, resulting in all of the aforementioned desirable characteristics of a universal unsupervised image outlier detector.
- In comparison to existing methods, our distance ensemble learning framework offers several advantages: (i) It achieves remarkable ranking accuracy and stability over various scenarios, as shown in Fig. 1; (ii) It is efficient, training-free, plug-and-play and can be seamlessly integrated with existing methods.

## 2 Related Work

### 2.1 Unsupervised Outlier Detection

Research in this field has branched into two primary categories: discrimination-based (e.g., one-class learners, statistics) and reconstruction-based (auto-encoder, GAN, self-supervised learning, etc.) approaches.

**Discrimination-based.** The discrimination-based methods refer to categorizing instances depending on some discriminative properties. Specifically, the one-class learning-based methods detect outliers by describing the inlier distribution (normality ) in the feature space. The representative methods are OC-SVM (Schölkopf et al., 2001), support vector data descriptor SVDD (Tax & Duin, 2004) and the deep version D-SVDD

(Ruff et al., 2018), which detects outliers by minimizing the volume of the hypersphere encompassing inliers. Besides, some classic methods discover outliers by examining the basic statistical characteristics of data, such as distance (Lin et al., 2021; 2022), density (Parzen, 1962; Ester et al., 1996; Breunig et al., 2000; Kriegel et al., 2008; Li et al., 2022), proximity (Ramaswamy et al., 2000; Angiulli & Pizzuti, 2002), etc.

**Reconstruction-based.** Deep outlier detectors commonly employ autoencoders or Generative Adversarial Networks (GANs) trained on inliers. During testing, samples that are not well-reconstructed are identified as anomalous (Hawkins et al., 2002; Sakurada & Yairi, 2014; Chen et al., 2017; Zhou & Paffenroth, 2017; Perera et al., 2019; Nguyen et al., 2019; Kim et al., 2020). Furthermore, deep generative models offer various techniques for anomaly detection (Schlegl et al., 2017; Deecke et al., 2018; Zenati et al., 2018; Schlegl et al., 2019). A recent trend in unsupervised outlier detection involves leveraging self-supervised learning to obtain more discriminative feature representations (Gidaris et al., 2018; Golan & El-Yaniv, 2018; Hendrycks et al., 2019; Tack et al., 2020; Sohn et al., 2021; Wang et al., 2020; Xu et al., 2023a).

### 2.2 Distance Metrics and Ensemble Learning

**Distance Metrics.** Distance transform is a necessary phase of outlier detection for computing outlier scores (likelihoods). Within deep reconstruction-based methods (Lai et al., 2019), the reconstruction error between a raw image and its reconstructed counterpart is calculated using classical metrics such as Cosine similarity and Euclidean distance. While distance transforms usually follow a multi-to-one/multi-protocol for statistical manifold learners, e.g., distance from instances to the target's mean. In such cases, distribution is a priority for leveraging some bilateral metrics, e.g., Bray-Curtis dissimilarity is widely used in bioinformatics to measure dissimilarities between ecological communities (Clarke et al., 2006).

**Ensemble Learning.** Ensemble learning is a popular topic in machine learning (Dong et al., 2020), such as bagging (Altman & Krzywinski, 2017), boosting (Chen & Guestrin, 2016) and stacking (Aboneh et al., 2022). One of the representative ensembling-based outlier detection methods is Isolation Forest (Liu et al., 2008) and its advanced versions (Zhao & Hryniewicki, 2018; Zhao et al., 2019; 2021; Xu et al., 2023a). However, their performance is limited in the high-dimensional space. Besides, the efficiency of traditional ensemble learning methods is also a significant problem. In this work, we are the first to clarify the ensembling objectives are not only the detection efficacy but also the stability over different dataset domains (Han et al., 2022) and outlier ratios (Camposeco et al., 2017; Wang et al., 2019b). To this end, we introduce a two-stage ensemble learning framework based on a careful selection of efficient distance transforms.

## 3 Building A Simple Baseline

### 3.1 Problem Definition

Given $n$ unlabelled image samples, we define the target dataset as $\mathbf{I} = \mathbf{I}_{\text{in}} \cup \mathbf{I}_{\text{out}}$, here $\mathbf{I}_{\text{in}}$ and $\mathbf{I}_{\text{out}}$ represent inliers and outliers, respectively. The outlier (contamination) ratio $\gamma \in (0, 1)$ is denoted as $\frac{\#\mathbf{I}_{\text{out}}}{\#\mathbf{I}_{\text{in}} + \#\mathbf{I}_{\text{out}}}$. Notably, this setup differs from the very related semi-supervised outlier detection task by not involving a prior *train-test split* for the target dataset. Typically, raw images are transformed to features $\mathbf{X} = \mathbf{X}_{\text{in}} \cup \mathbf{X}_{\text{out}} = \{\mathbf{x}_i\}_{i=1}^{n}$, here $i$ is the feature index and $n$ refers to the amount of target dataset. The primary goal of unsupervised image outlier detection is to develop an outlier score function $\text{F}(\cdot)$ to assess the outlier likelihood of each $\mathbf{x}_i$:

$$\text{Y}(\mathbf{x}_i) = \begin{cases} 1, & \text{if } \text{F}(\mathbf{x}_i) \geq \tau; \\ 0, & \text{otherwise,} \end{cases} \tag{1}$$

where $\text{Y} = 1$ (outlier) and $\text{Y} = 0$ (inlier) refers to the predicted labels and $\tau$ is the threshold (decision boundary). In this study, we not only measure the ranking accuracy of $\text{F}(\cdot)$, aligning with the related works (Ruff et al., 2018; Lai et al., 2019; Lin et al., 2021; 2022), but also evaluate the classification accuracy of $\tau$.

### 3.2 Understanding the High-dimensional Space

Unsupervised image outlier detection refers to detecting outliers in the high-dimensional space. Deviated from the basic intuition in the 2/3-$d$ space, the high-dimensional data lies in a $(d, r)$-hypersphere, where $d$ is

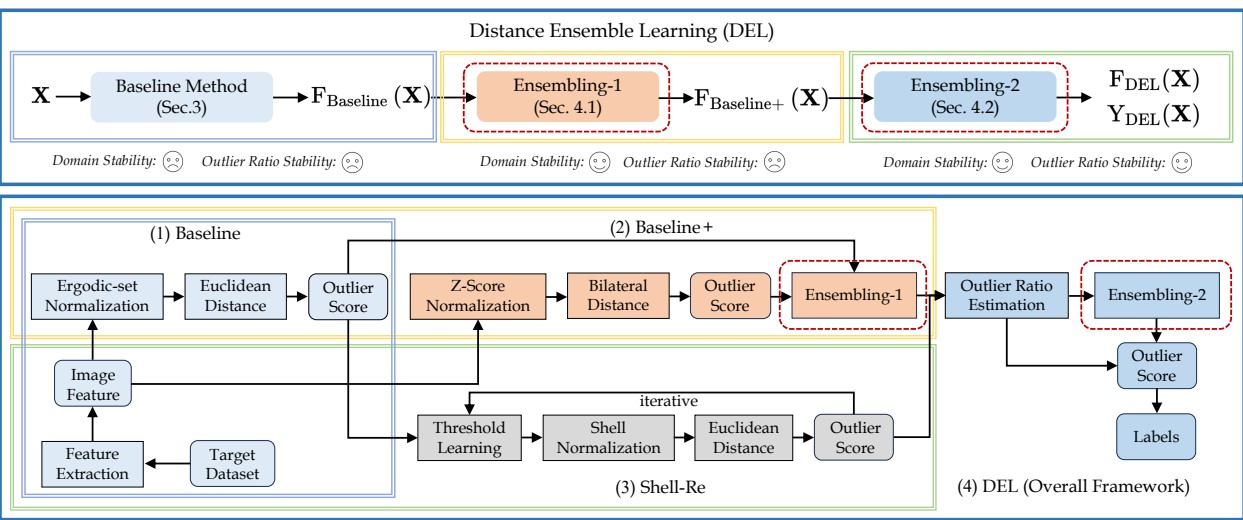

Figure 2: The overview of our DEL framework. The upper plot illustrates DEL's overall pipeline, while the lower plot provides detailed insights. Module (4): DEL comprises three modules with two ensembling stages. Module (1) serves as our baseline: Ergodic-set normalized Euclidean distance. Given the baseline's susceptibility to domain variation, we enhance it with Z-score normalized bilateral distance in module (2). Additionally, we ensemble it with Shell-Re (module (3)), ensuring stability across outlier ratios.

the dimension and $r$ is the radius. The volume of $d$-dimensional hypersphere $\mathrm{V}_d(r)$ is defined as:

$$\mathrm{V}_d(r) = \frac{\pi^{d/2} r^d}{\Gamma(1 + d/2)}, \tag{2}$$

where $\Gamma(\cdot)$ is the Gamma function. For all inlier samples $\mathbf{X}_{\mathrm{in}}$ in a certain hypersphere, if we shrink its radius $r$ with a small positive value $\epsilon$, the volume ratio will satisfy:

$$\lim_{d \to \infty} \frac{\mathrm{V}_d((1 - \epsilon)r)}{\mathrm{V}_d(r)} = \lim_{d \to \infty} (1 - \epsilon)^d \leq \lim_{d \to \infty} e^{-\epsilon d} = 0. \tag{3}$$

If $\epsilon$ is fixed and $d \to \infty$, the volume inside the hypersphere rapidly tends to zero, i.e., almost all of the volume is concentrated on the hypersphere's surface (Hopcroft & Kannan, 2014). This motivates us that the distance from each sample to the centroid of inliers will be a naturally effective outlier score since:

$$||\mathbf{X}_{\mathrm{in}} - \mathbf{m}_{\mathbf{X}_{\mathrm{in}}}||_2 \overset{a.s.}{=} r, \;\; ||\mathbf{X}_{\mathrm{out}} - \mathbf{m}_{\mathbf{X}_{\mathrm{in}}}||_2 > r, \tag{4}$$

where $|| \cdot ||_2$ is $\ell_2$-norm (Euclidean distance) and $\overset{a.s.}{=}$ refers to the almost-sure equality.

### 3.3 A Baseline Method

In scenarios where the outlier ratio $\gamma$ is relatively low, inliers will predominantly make up the target dataset. Hence, the centroid of the inliers, $\mathbf{m}_{\mathbf{X}_{\mathrm{in}}}$ can be replaced with the mean of the target dataset's features $\mathbf{m}_{\mathbf{X}}$:

$$\mathbf{m}_{\mathbf{X}} = [\mathbf{m}_1, \cdots, \mathbf{m}_j, \cdots, \mathbf{m}_d]^\top, \mathbf{m}_{j \in [1,d]} = \frac{1}{n} \sum_{i=1}^{n} \mathbf{x}_{i,j}, \tag{5}$$

where $j$ is the index of dimension. Inspired by Shell Theory (Lin et al., 2021), the normalization procedure is critical for distance computation with image features. Here, we adopt the Ergodic-set normalization E-norm$(\cdot)$ (Lin et al., 2022), which is specifically designed for unsupervised image outlier detection tasks, illustrated as:

$$\mathrm{E\text{-}norm}(\mathbf{X}) = \frac{\mathbf{X} - \mathbf{v}_{\mathrm{E}}}{||\mathbf{X} - \mathbf{v}_{\mathrm{E}}||_2}, \mathbf{v}_{\mathrm{E}} = \frac{1}{n \cdot d} \sum_{i=1}^{n} \sum_{j=1}^{d} \mathbf{x}_{i,j}, \tag{6}$$

where $\mathbf{v}_E$ is the corresponding reference scalar (vector with the same value in each dimension). Such that, the outlier score function of our baseline method is formulated as:

$$F_{\text{Baseline}}(\mathbf{X}) = ||\text{E-norm}(\mathbf{X}) - \text{E-norm}(\mathbf{m_X})||_2. \tag{7}$$

## 4 Distance Ensemble Learning

Despite the baseline method (Eq. 7) described above still suffering from the inevitable instability over various dataset domains and outlier ratios, as it is simple and intuitive, we can improve its stability step by step in an explainable way. In Sec. 4.1, we improve its stability across various target dataset domains by ensembling with a conditional bilateral distance. In Sec. 4.2, we further integrate the advanced baseline with a high outlier ratio-specific method, to address the instability of varying outlier ratios. Details are shown in Fig. 2.

### 4.1 Improving the Domain Stability

**Motivation.** For outlier detection and some related tasks, Euclidean distance/Cosine similarity is a commonly used distance metric (Lai et al., 2019; Lin et al., 2021; 2023). But as a symmetric metric, it lacks separability for some specific scenarios, e.g., if $\mathbf{x}_1$ and $\mathbf{x}_2$ belong to inlier and outlier respectively, and $||\mathbf{x}_1 - \mathbf{x}_2||_2$ is relative large, however, $\mathbf{x}_1$ and $\mathbf{x}_2$ may be about the $\mathbf{m_X}$ symmetry, i.e., $||\mathbf{x}_1 - \mathbf{m_X}||_2 \overset{a.s.}{=} ||\mathbf{x}_2 - \mathbf{m_X}||_2$, such that the Euclidean distance will mistakenly predict two similar outlier scores. To address the issue caused by symmetric metrics, we introduce an asymmetric bilateral distance metric, to integrate with the baseline.

**Conditional Bilateral Distance.** The bilateral distance metric is formulated as:

$$F_{\text{Bilateral}}(\mathbf{X}) = \frac{||\text{Z-norm}(\mathbf{X}) - \text{Z-norm}(\mathbf{m_X})||_1}{||\text{Z-norm}(\mathbf{X}) + \text{Z-norm}(\mathbf{m_X})||_1}, \tag{8}$$

where $||\cdot||_1$ refers to $\ell_1$-norm (Manhatten distance) and Z-norm($\cdot$) is the Z-score normalization with the same reference scalar $\mathbf{v}_E$ in Ergodic-set normalization, formulated as:

$$\text{Z-norm}(\mathbf{X}) = \frac{\mathbf{X} - \mathbf{v}_E}{\sqrt{\sigma^2}}, \sigma^2 = \frac{1}{n \cdot d} \sum_{i=1}^{n} \sum_{j=1}^{d} (\mathbf{x}_{i,j} - \mathbf{v}_E[j])^2. \tag{9}$$

The related version of this bilateral distance is somewhat termed as Bray-Curtis dissimilarity is widely utilized in the fields of bioinformatics (Clarke et al., 2006). The empirical evidence has demonstrated its superiority over Euclidean distance and Cosine similarity in such areas. Notably, this is its first usage for unsupervised image outlier detection. So a comprehensive analysis both theoretically and empirically is crucial.

**Theoretical Analysis.** We first decompose the bilateral distance (Eq. 8) into two separate components: the numerator $N_1 = ||\text{Z-norm}(\mathbf{X}) - \text{Z-norm}(\mathbf{m_X})||_1$ and denominator $N_2 = ||\text{Z-norm}(\mathbf{X}) + \text{Z-norm}(\mathbf{m_X})||_1$.

**Lemma 1.** *Given any* $a, b \in \mathbb{R}, |a - b| + |a + b| = 2 \cdot \max\{|a|, |b|\}.$

*Proof.* Shown in the Appendix. □

**Theorem 1.** $N_1$ *and* $N_2$ *are* *conditionally symmetric*.

*Proof.* For $\forall \mathbf{x}_i \in \mathbf{X}$, the value of each dimension $j \in [1, \cdots, d]$ adheres to the subsequent equation, as established by Lemma 1:

$$\frac{(|\mathbf{x}_{i,j} - \mathbf{m_X}[j]|) + (|\mathbf{x}_{i,j} + \mathbf{m_X}[j]|)}{2} = \max\{|\mathbf{x}_{i,j}|, |\mathbf{m_X}[j]|\}, \tag{10}$$

where $|\cdot|$ refers to the absolute operation and Eq. 10 can be extended to encompass all dimensions:

$$\frac{||\mathbf{x}_i - \mathbf{m_X}||_1 + ||\mathbf{x}_i + \mathbf{m_X}||_1}{2} = \max\{||\mathbf{x}_i - \mathbf{o}||_1, ||\mathbf{m_X} - \mathbf{o}||_1\}, \tag{11}$$

where $\mathbf{o} = [0_1, \cdots, 0_d]$ represents the origin. For simplification of the right-hand side of Eq. 10, Z-score normalization Z-norm($\cdot$) is employed to obtain the following conclusion:

$$\frac{||\text{Z-norm}(\mathbf{X}) - \text{Z-norm}(\mathbf{m_X})||_1 + ||\text{Z-norm}(\mathbf{X}) + \text{Z-norm}(\mathbf{m_X})||_1}{2} = ||\text{Z-norm}(\mathbf{X}) - \mathbf{o}||_1. \tag{12}$$

If $||\text{Z-norm}(\mathbf{X}) - \mathbf{o}||_1$ is a constant and $||\text{Z-norm}(\mathbf{X}) - \mathbf{o}||_1 \geq \max(||\text{Z-norm}(\mathbf{X}) - \text{Z-norm}(\mathbf{m_X})||_1)$, $N_1$ and $N_2$ will be independently symmetric with each other. $\qquad\square$

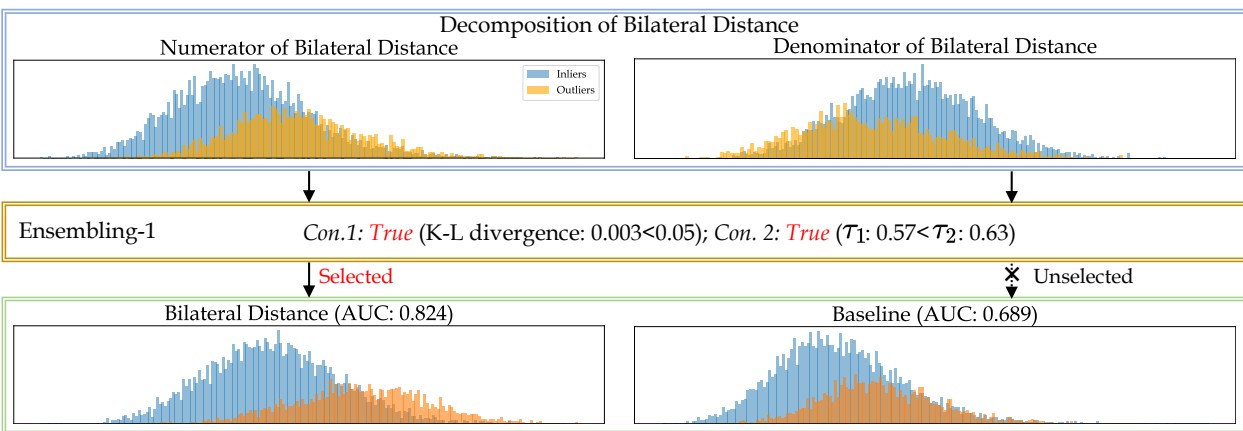

Figure 3: A visualization example of the ensembling-1. In the upper plot, the numerator and denominator exhibit near-symmetry, prompting the validation of conditional bilateral distance, as shown in the lower plot.

**Conditional Adoption.** Ideally, outliers are located at a greater $\ell_1$-norm value from the mean Z-norm($\mathbf{m_X}$) compared to any inlier, i.e., $\max\{||\text{Z-norm}(\mathbf{X}_{\text{in}}) - \text{Z-norm}(\mathbf{m_X})||_1\} < \min\{||\text{Z-norm}(\mathbf{X}_{\text{out}}) - \text{Z-norm}(\mathbf{m_X})||_1\}$. Leveraging its self-symmetry, the bilateral distance adheres to $\min\{||\text{Z-norm}(\mathbf{X}_{\text{in}}) + \text{Z-norm}(\mathbf{m_X})||_1\} > \max\{||\text{Z-norm}(\mathbf{X}_{\text{out}}) + \text{Z-norm}(\mathbf{m_X})||_1\}$, thereby the gap between inliers and outliers will be enhanced. Considering the conditional symmetry, we now discuss its practical adoption. In reality, $||\text{Z-norm}(\mathbf{X}) - \mathbf{o}||_1$ often exhibits a Gaussian-like distribution rather than a constant. Hence, we focus on directly assessing the symmetry between $N_1$ and $N_2$. To further separate the inliers and outliers, $N_1$ and $N_2$ should satisfy both:

*Condition 1.* The distribution of $N_1$ exhibits similarity to the distribution of $N_2$.

To measure the similarity, we utilize K-L divergence, where we constrain its value as less than **0.05**.

$$D_{\text{KL}}(N_1||N_2) = \sum_{i=1}^{n} N_1(\mathbf{x}_i) \log\left(\frac{N_1(\mathbf{x}_i)}{N_2(\mathbf{x}_i)}\right). \tag{13}$$

*Condition 2.* $N_1$ and $N_2$ are independent of each other.

In statistics, the "3-sigma" rule (Pukelsheim, 1994) suggests data points that are more than three standard deviations $\sigma$ from the mean $\mu$ can be considered as out-of-distribution instances. So we test whether: $\tau_1 < \tau_2$, here $\tau_1 = \mu(N_1) + 3 \cdot \sigma(N_1), \tau_2 = \mu(N_2) - 3 \cdot \sigma(N_2)$. Besides, it constrains the symmetry to the right, i.e., the majority of $N_2$ is larger than $N_1$.

**Ensembling-1.** If Con. 1 and Con. 2 are all satisfied, and the bilateral distance will be selected. Otherwise, the baseline remains as before. The *advanced* baseline $F_{\text{Baseline+}}$ is formulated as:

$$F_{\text{Baseline+}}(\mathbf{X}) = \begin{cases} F_{\text{Bilateral}}(\mathbf{X}), & \text{if Con.1 \& 2 are True;} \\ F_{\text{Baseline}}(\mathbf{X}), & \text{otherwise.} \end{cases} \tag{14}$$

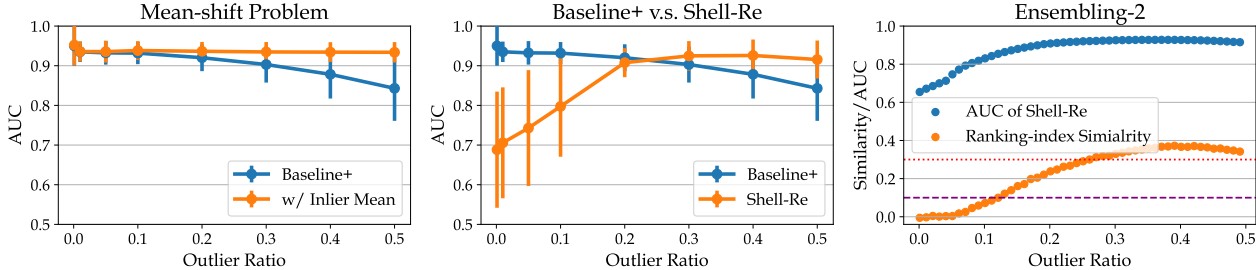

Figure 4: The left plot illustrates the decrease in efficacy of the advanced baseline, referred to as the mean-shift problem. The middle plot compares our advanced baseline with Shell-Re, a high-ratio specific outlier detector. The right plot demonstrates the implicit estimation of the outlier ratio via the ranking index similarity.

## 4.2 Improving the Outlier Ratio Stability

**Mean-shift Problem.** If the outlier ratio $\gamma$ is high, the mean of target dataset $\mathbf{m_X}$ encounters a significant shift, i.e., the gap between $\mathbf{m_X}$ and the mean of inliers $\mathbf{m_{X_{in}}}$ becomes larger. Thus, the efficacy of the advanced baseline suffers from a decrease at high outlier ratios, demonstrated in Fig. 4 (left plot). As some invariant-mean searching algorithms using such kernel density estimation is time-consuming (Wu et al., 2015), we convert the view that directly integrates the advanced baseline with a high-$\gamma$ specific outlier detector.

**Revisiting Shell-Re.** Shell-Re derives from Shell Theory (Lin et al., 2021), a high-dimensional theory that illustrate that the image feature representation can be dramatically improved with ideal Shell normalization:

$$\text{S-norm}(\mathbf{X}) = \frac{\mathbf{X} - \mathbf{v}_{\text{S}}(\mathbf{X}_{\text{out}})}{||\mathbf{X} - \mathbf{v}_{\text{S}}(\mathbf{X}_{\text{out}})||_2}, \mathbf{v}_{\text{S}}(\mathbf{X}_{\text{out}}) = \left[\frac{1}{n}\sum_{i=1}^{n}\mathbf{X}_{\text{out}}[i][1], \cdots, \frac{1}{n}\sum_{i=1}^{n}\mathbf{X}_{\text{out}}[i][d]\right], \quad (15)$$

where $\mathbf{v}_{\text{S}}$ is its corresponding reference vector and $\mathbf{X}_{\text{out}}$ refers to ground-truth outliers. To be consistent with our advanced baseline while aligning with the Guassian-distribution assumption in the original paper, we take $\text{F}_{\text{Baseline}}(\mathbf{X})$ (Eq. 7) as a reliable initial outlier score foundation. Subsequently, the algorithm aims to identify potential outliers, denoted as $\mathbf{X}'_{\text{out}}$, using a classic threshold learning method Median Absolute Deviation, i.e., MAD (Rousseeuw & Croux, 1993) under the Robust-Least-Square (RLS) paradigm, so the predicted outlier candidates at the first iteration are $\mathbf{X}'_{\text{out}} = \{\mathbf{x}_i | \text{F}_{\text{Baseline}}(\mathbf{x}_i) > \text{Median} + k \cdot \text{MAD}\}$[2]. If the converged outlier prediction $\mathbf{X}^*_{\text{out}} \approx \mathbf{X}_{\text{out}}$, the outlier score function of Shell-Re will demonstrate excellent separability:

$$\text{F}_{\text{Shell-Re}}(\mathbf{X}) = ||\text{S-norm}(\mathbf{X}, \mathbf{v}_{\text{S}}(\mathbf{X}^*_{\text{out}})) - \text{S-norm}(\mathbf{m_X}, \mathbf{v}_{\text{S}}(\mathbf{X}^*_{\text{out}}))||_2. \quad (16)$$

Since the predicted threshold is only effective with high outlier ratios (Lin et al., 2021), the detection accuracy of Shell-Re is positively correlated with $\gamma$, as shown in Fig. 4 (middle plot). Besides, as threshold learning across different outlier ratios is often ill-conditioned (Cherapanamjeri et al., 2017), the *ideal* ensembling is:

$$\text{F}_{\text{Ideal}}(\mathbf{X}) = \begin{cases} \text{F}_{\text{Shell-Re}}(\mathbf{X}), & \text{if } \gamma \text{ is high;} \\ \\ \text{F}_{\text{Baseline+}}(\mathbf{X}), & \text{otherwise.} \end{cases} \quad (17)$$

Recognizing that a known $\gamma$ prior is impractical in real-world applications, we introduce an implicit $\gamma$-estimator.

**Outlier Ratio Estimation.** For any given target dataset $\mathbf{X}$, we initially apply both advanced baseline and Shell-Re simultaneously, resulting in two lists of outlier scores $\text{F}_{\text{Baseline+}}(\mathbf{X})$ and $\text{F}_{\text{Shell-Re}}(\mathbf{X})$. Subsequently, they are arranged in ascending order, and we denote the resulting ranking-index lists as $\text{R}_{\text{Baseline+}}$ and $\text{R}_{\text{Shell-Re}}$, respectively. The outlier ratio $\gamma$ can be approximately estimated by analyzing the similarity between $\text{R}_{\text{Baseline+}}$ and $\text{R}_{\text{Shell-Re}}$ with the following three reasons: (i) As the Ergodic-set normalization (Eq. 6) is independent with $\gamma$, the advanced baseline $\text{F}_{\text{Baseline+}}(\mathbf{X})$ serves as a reliable reference over a wide range of outlier ratios. (ii) When $\gamma$ is low, Shell-Re tends to underperform, resulting in low similarity. Conversely,

---

[2]In this work, we set $k = 1$ that is more robust to higher outlier ratio than the default $k = 2$ in the original paper.

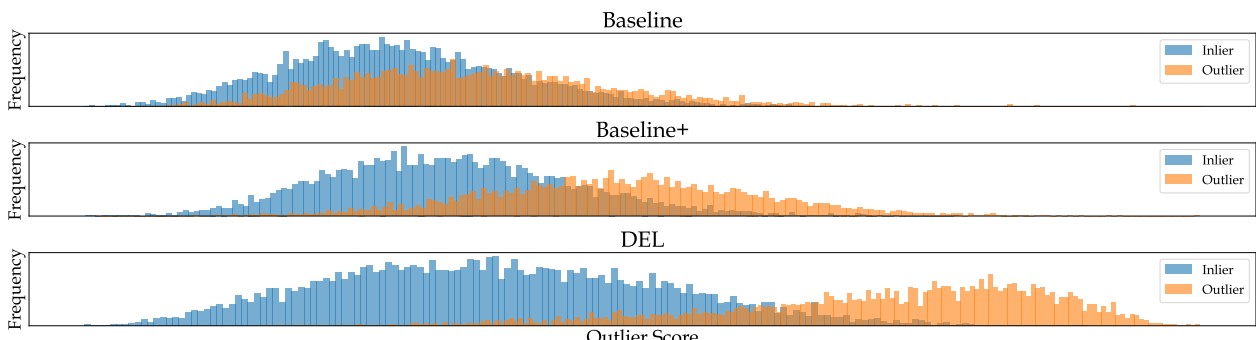

Figure 5: Histogram of outlier scores generated by the baseline, advanced baseline, and DEL, respectively, for an example conducted on CIFAR-10 (Dog) while the outlier ratio is 0.4. It is evident that the two ensembling modules significantly contribute to the separability between the inlier and outlier distributions.

with a high $\gamma$, Shell-Re merely refines the ranking of a small number of outliers, indicative of high similarity. (iii) The advanced baseline and Shell-Re share a consistent structure, i.e., the distance transform relative to the mean of the target dataset, making $R_{\text{Baseline+}}$ and $R_{\text{Shell-Re}}$ comparable. Here, we consider the Spearman rank-order correlation coefficient $\phi \in [-1, 1]$ (Fieller & Pearson, 1961) to measure the ranking-index similarity:

$$\phi = 1 - \frac{6 \sum_{i=1}^{n} (R_{\text{Baseline+}}[i] - R_{\text{Shell-Re}}[i])^2}{n\,(n^2 - 1)}. \tag{18}$$

As shown in Fig. 4 (right plot), the effectiveness of Shell-Re is positively correlated with the ranking-index similarity. Thus, $\phi$ can be considered as an approximate $\gamma$.

**Ensembling-2.** To maintain the simplicity, we denote $\phi_1, \phi_2$, representing low-ratio and high-ratio boundaries, respectively. The outlier score function of the overall DEL framework is:

$$F_{\text{DEL}}(\mathbf{X}) = \begin{cases} F_{\text{Shell-Re}}(\mathbf{X})\,, & \text{if } \phi > \phi_2; \\ \widehat{F}_{\text{Baseline+}}(\mathbf{X}) + \widehat{F}_{\text{Shell-Re}}(\mathbf{X})\,, & \text{else if } \phi_1 \le \phi \le \phi_2; \\ F_{\text{Baseline+}}(\mathbf{X})\,, & \text{otherwise.} \end{cases} \tag{19}$$

where $\widehat{F}(\cdot)$ refers to the Min-max normalized outlier score function and $\phi_1, \phi_2$ are fixed to 0.1 and 0.3, respectively. As shown in Fig. 5, the two ensembling stages progressively improve the performance. More importantly, DEL is an inherent binary classifier, since the absolute value of the estimated outlier ratio $|\phi|$ can be inherently considered a threshold (decision boundary) $\tau$, i.e.,

$$Y_{\text{DEL}}(\mathbf{X}) = F_{\text{DEL}}(\mathbf{X}, |\phi|) = \begin{cases} 1, & \text{if } F_{\text{DEL}}(\mathbf{x}_i) \ge \text{sort}(F_{\text{DEL}})[n \cdot (1 - |\phi|)]; \\ 0, & \text{otherwise.} \end{cases} \tag{20}$$

where $\text{sort}(\cdot)$ refers to the ascending sort operation. Additionally, as most of the proposed methods (Lai et al., 2019; Xu et al., 2023a) suffer from the efficacy decrease with high outlier ratios while the estimated outlier ratio is model-agnostic, DEL can be seamlessly integrated with any low-ratio effective method $M(\cdot)$:

$$F_{\text{M+DEL}}(\mathbf{X}) = \begin{cases} F_{\text{Shell-Re}}(\mathbf{X})\,, & \text{if } \phi > \phi_2; \\ \widehat{F}_{\text{M}}(\mathbf{X}) + \widehat{F}_{\text{Shell-Re}}(\mathbf{X})\,, & \text{else if } \phi_1 \le \phi \le \phi_2;\ \ Y_{\text{M+DEL}}(\mathbf{X}) = F_{\text{M+DEL}}(\mathbf{X}, |\phi|). \\ F_{\text{M}}(\mathbf{X})\,, & \text{otherwise.} \end{cases} \tag{21}$$

## 5 Experiments

### 5.1 Settings

**Target Datasets.** Building upon previous related research (Lin et al., 2021; 2022), we leverage the raw pixel features of gray-scale datasets, including MNIST (LeCun & Cortes, 2010) and Fashion-MNIST (Xiao et al.,

Table 1: Average results over a wide range of outlier ratios ($[0.01, \cdots, 0.5]$) with SOTAs on a series of RGB datasets. Red and Blue indicates the best and second-best results, respectively.

| Feat. | Method | Venue | STL-10 AUC | PR-I | PR-O | Internet AUC | PR-I | PR-O | Caltech101 AUC | PR-I | PR-O | CIFAR10 AUC | PR-I | PR-O | CIFAR100 AUC | PR-I | PR-O | MIT-Places-Small AUC | PR-I | PR-O |
|---|---|---|---|---|---|---|---|---|---|---|---|---|---|---|---|---|---|---|---|---|
| ResNet-50 | OC-SVM | NeuCom-01 | 0.902 | 0.934 | 0.771 | 0.935 | 0.955 | 0.844 | 0.965 | 0.974 | 0.934 | 0.802 | 0.878 | 0.561 | 0.858 | 0.910 | 0.662 | 0.805 | 0.885 | 0.574 |
| | IF | ICDM-08 | 0.818 | 0.902 | 0.557 | 0.848 | 0.920 | 0.595 | 0.896 | 0.941 | 0.754 | 0.760 | 0.861 | 0.498 | 0.784 | 0.873 | 0.532 | 0.671 | 0.826 | 0.391 |
| | INNE | ComInt-18 | 0.846 | 0.922 | 0.553 | 0.862 | 0.935 | 0.581 | 0.891 | 0.948 | 0.726 | 0.787 | 0.885 | 0.471 | 0.867 | 0.928 | 0.629 | 0.761 | 0.873 | 0.450 |
| | D-SVDD | ICML-18 | 0.601 | 0.777 | 0.378 | 0.695 | 0.821 | 0.484 | 0.785 | 0.855 | 0.664 | 0.539 | 0.762 | 0.293 | 0.567 | 0.771 | 0.356 | 0.584 | 0.777 | 0.344 |
| | RSRAE | ICLR-19 | 0.891 | 0.931 | 0.740 | 0.935 | 0.960 | 0.821 | 0.989 | 0.995 | 0.961 | 0.778 | 0.865 | 0.532 | 0.893 | 0.944 | 0.696 | 0.735 | 0.843 | 0.497 |
| | GOAD | ICLR-20 | 0.942 | 0.967 | 0.819 | 0.973 | 0.986 | 0.888 | 0.985 | 0.993 | 0.943 | 0.835 | 0.904 | 0.603 | 0.897 | 0.941 | 0.709 | 0.833 | 0.905 | 0.608 |
| | RCA | IJCAI-21 | 0.497 | 0.748 | 0.252 | 0.502 | 0.749 | 0.253 | 0.503 | 0.753 | 0.266 | 0.499 | 0.749 | 0.252 | 0.494 | 0.748 | 0.252 | 0.490 | 0.748 | 0.252 |
| | Shell-Re | TPAMI-21 | 0.870 | 0.980 | 0.669 | 0.916 | 0.986 | 0.701 | 0.916 | 0.986 | 0.701 | 0.860 | 0.958 | 0.640 | 0.846 | 0.959 | 0.592 | 0.759 | 0.896 | 0.541 |
| | NeuTraL | ICML-21 | 0.839 | 0.904 | 0.648 | 0.899 | 0.947 | 0.750 | 0.802 | 0.874 | 0.689 | 0.744 | 0.854 | 0.479 | 0.793 | 0.906 | 0.550 | 0.718 | 0.830 | 0.443 |
| | ICL | ICLR-22 | 0.920 | 0.952 | 0.767 | 0.957 | 0.979 | 0.837 | 0.966 | 0.984 | 0.896 | 0.827 | 0.897 | 0.583 | 0.904 | 0.948 | 0.732 | 0.795 | 0.882 | 0.527 |
| | LUNAR | AAAI-22 | 0.759 | 0.854 | 0.494 | 0.770 | 0.872 | 0.505 | 0.883 | 0.921 | 0.727 | 0.748 | 0.847 | 0.481 | 0.825 | 0.881 | 0.608 | 0.655 | 0.792 | 0.378 |
| | ECOD | TKDE-22 | 0.892 | 0.944 | 0.677 | 0.889 | 0.947 | 0.654 | 0.964 | 0.981 | 0.889 | 0.856 | 0.917 | 0.630 | 0.906 | 0.949 | 0.734 | 0.765 | 0.865 | 0.476 |
| | LVAD | ECCV-22 | 0.940 | 0.966 | 0.816 | 0.958 | 0.978 | 0.860 | 0.978 | 0.988 | 0.935 | 0.832 | 0.904 | 0.600 | 0.910 | 0.950 | 0.737 | 0.820 | 0.896 | 0.600 |
| | SLAD | ICML-23 | 0.920 | 0.956 | 0.740 | 0.951 | 0.973 | 0.817 | 0.860 | 0.932 | 0.693 | 0.841 | 0.911 | 0.592 | 0.904 | 0.946 | 0.720 | 0.818 | 0.894 | 0.566 |
| | DeepIF | TKDE-23 | 0.807 | 0.896 | 0.543 | 0.828 | 0.910 | 0.577 | 0.812 | 0.910 | 0.605 | 0.749 | 0.865 | 0.466 | 0.774 | 0.886 | 0.495 | 0.700 | 0.838 | 0.396 |
| | DEL | — | 0.972 | 0.989 | 0.869 | 0.978 | 0.990 | 0.883 | 0.985 | 0.995 | 0.939 | 0.927 | 0.964 | 0.766 | 0.941 | 0.972 | 0.813 | 0.877 | 0.931 | 0.680 |
| CLIP | OC-SVM | NeuCom-01 | 0.896 | 0.925 | 0.780 | 0.870 | 0.920 | 0.648 | 0.945 | 0.963 | 0.896 | 0.846 | 0.903 | 0.630 | 0.832 | 0.893 | 0.634 | 0.842 | 0.901 | 0.557 |
| | IF | ICDM-08 | 0.930 | 0.955 | 0.812 | 0.901 | 0.944 | 0.680 | 0.969 | 0.985 | 0.906 | 0.878 | 0.928 | 0.667 | 0.854 | 0.913 | 0.640 | 0.852 | 0.917 | 0.583 |
| | INNE | ComInt-18 | 0.859 | 0.926 | 0.580 | 0.797 | 0.903 | 0.430 | 0.896 | 0.952 | 0.731 | 0.839 | 0.917 | 0.521 | 0.850 | 0.911 | 0.639 | 0.798 | 0.908 | 0.399 |
| | D-SVDD | ICML-18 | 0.585 | 0.791 | 0.319 | 0.653 | 0.838 | 0.332 | 0.704 | 0.857 | 0.428 | 0.527 | 0.760 | 0.289 | 0.583 | 0.808 | 0.267 | 0.495 | 0.765 | 0.291 |
| | RSRAE | ICLR-19 | 0.915 | 0.947 | 0.795 | 0.897 | 0.939 | 0.679 | 0.983 | 0.992 | 0.959 | 0.855 | 0.914 | 0.641 | 0.885 | 0.931 | 0.707 | 0.821 | 0.898 | 0.493 |
| | GOAD | ICLR-20 | 0.949 | 0.968 | 0.856 | 0.918 | 0.954 | 0.718 | 0.977 | 0.989 | 0.938 | 0.890 | 0.936 | 0.695 | 0.876 | 0.926 | 0.687 | 0.909 | 0.948 | 0.673 |
| | RCA | IJCAI-21 | 0.957 | 0.974 | 0.866 | 0.932 | 0.964 | 0.746 | 0.980 | 0.991 | 0.949 | 0.893 | 0.937 | 0.702 | 0.876 | 0.922 | 0.695 | 0.914 | 0.951 | 0.681 |
| | Shell-Re | TPAMI-21 | 0.861 | 0.981 | 0.648 | 0.876 | 0.986 | 0.670 | 0.893 | 0.985 | 0.700 | 0.874 | 0.969 | 0.605 | 0.821 | 0.952 | 0.552 | 0.885 | 0.965 | 0.666 |
| | NeuTraL | ICML-21 | 0.818 | 0.870 | 0.590 | 0.854 | 0.926 | 0.557 | 0.464 | 0.725 | 0.369 | 0.751 | 0.846 | 0.474 | 0.837 | 0.905 | 0.605 | 0.705 | 0.824 | 0.431 |
| | ICL | ICLR-22 | 0.938 | 0.964 | 0.807 | 0.917 | 0.957 | 0.703 | 0.961 | 0.982 | 0.887 | 0.890 | 0.940 | 0.659 | 0.918 | 0.954 | 0.758 | 0.869 | 0.931 | 0.565 |
| | LUNAR | AAAI-22 | 0.802 | 0.861 | 0.643 | 0.781 | 0.861 | 0.547 | 0.963 | 0.966 | 0.940 | 0.785 | 0.863 | 0.532 | 0.894 | 0.927 | 0.772 | 0.741 | 0.841 | 0.437 |
| | ECOD | TKDE-22 | 0.972 | 0.982 | 0.912 | 0.961 | 0.977 | 0.879 | 0.988 | 0.994 | 0.970 | 0.921 | 0.951 | 0.768 | 0.905 | 0.939 | 0.758 | 0.927 | 0.960 | 0.724 |
| | LVAD | ECCV-22 | 0.956 | 0.974 | 0.868 | 0.889 | 0.933 | 0.678 | 0.975 | 0.988 | 0.937 | 0.896 | 0.938 | 0.712 | 0.903 | 0.940 | 0.753 | 0.899 | 0.946 | 0.629 |
| | SLAD | ICML-23 | 0.925 | 0.950 | 0.808 | 0.888 | 0.936 | 0.641 | 0.916 | 0.955 | 0.804 | 0.882 | 0.930 | 0.674 | 0.895 | 0.938 | 0.729 | 0.875 | 0.935 | 0.570 |
| | DeepIF | TKDE-23 | 0.910 | 0.946 | 0.751 | 0.903 | 0.946 | 0.701 | 0.936 | 0.970 | 0.812 | 0.853 | 0.915 | 0.621 | 0.842 | 0.911 | 0.616 | 0.830 | 0.911 | 0.528 |
| | DEL | — | 0.988 | 0.995 | 0.928 | 0.985 | 0.994 | 0.916 | 0.981 | 0.992 | 0.954 | 0.957 | 0.981 | 0.808 | 0.954 | 0.980 | 0.827 | 0.979 | 0.991 | 0.862 |
| Average | OC-SVM | NeuCom-01 | 0.899 | 0.930 | 0.776 | 0.903 | 0.937 | 0.746 | 0.955 | 0.968 | 0.915 | 0.824 | 0.891 | 0.595 | 0.845 | 0.901 | 0.648 | 0.824 | 0.893 | 0.565 |
| | IF | ICDM-08 | 0.874 | 0.928 | 0.685 | 0.874 | 0.932 | 0.638 | 0.932 | 0.963 | 0.830 | 0.819 | 0.895 | 0.583 | 0.819 | 0.893 | 0.586 | 0.762 | 0.872 | 0.487 |
| | INNE | ComInt-18 | 0.852 | 0.924 | 0.567 | 0.829 | 0.919 | 0.505 | 0.894 | 0.950 | 0.728 | 0.813 | 0.901 | 0.496 | 0.859 | 0.919 | 0.634 | 0.780 | 0.890 | 0.425 |
| | D-SVDD | ICML-18 | 0.593 | 0.784 | 0.349 | 0.674 | 0.829 | 0.408 | 0.745 | 0.856 | 0.546 | 0.533 | 0.761 | 0.291 | 0.575 | 0.790 | 0.311 | 0.539 | 0.771 | 0.317 |
| | RSRAE | ICLR-19 | 0.903 | 0.939 | 0.767 | 0.916 | 0.950 | 0.750 | 0.986 | 0.993 | 0.960 | 0.816 | 0.890 | 0.587 | 0.889 | 0.938 | 0.702 | 0.778 | 0.870 | 0.495 |
| | GOAD | ICLR-20 | 0.946 | 0.968 | 0.838 | 0.945 | 0.970 | 0.803 | 0.981 | 0.991 | 0.941 | 0.862 | 0.920 | 0.649 | 0.886 | 0.934 | 0.698 | 0.871 | 0.926 | 0.640 |
| | RCA | IJCAI-21 | 0.727 | 0.861 | 0.559 | 0.717 | 0.857 | 0.499 | 0.742 | 0.872 | 0.608 | 0.696 | 0.843 | 0.477 | 0.685 | 0.835 | 0.473 | 0.702 | 0.850 | 0.466 |
| | Shell-Re | TPAMI-21 | 0.866 | 0.981 | 0.659 | 0.896 | 0.986 | 0.686 | 0.905 | 0.986 | 0.701 | 0.867 | 0.964 | 0.623 | 0.834 | 0.956 | 0.572 | 0.822 | 0.931 | 0.604 |
| | NeuTraL | ICML-21 | 0.828 | 0.887 | 0.619 | 0.877 | 0.937 | 0.653 | 0.633 | 0.799 | 0.529 | 0.748 | 0.850 | 0.476 | 0.815 | 0.905 | 0.577 | 0.711 | 0.827 | 0.437 |
| | ICL | ICLR-22 | 0.929 | 0.958 | 0.787 | 0.937 | 0.968 | 0.770 | 0.964 | 0.983 | 0.891 | 0.859 | 0.918 | 0.621 | 0.911 | 0.951 | 0.745 | 0.832 | 0.906 | 0.546 |
| | LUNAR | AAAI-22 | 0.781 | 0.857 | 0.568 | 0.776 | 0.867 | 0.526 | 0.923 | 0.944 | 0.834 | 0.766 | 0.855 | 0.507 | 0.859 | 0.904 | 0.690 | 0.698 | 0.817 | 0.408 |
| | ECOD | TKDE-22 | 0.932 | 0.963 | 0.795 | 0.925 | 0.962 | 0.766 | 0.976 | 0.987 | 0.930 | 0.888 | 0.934 | 0.699 | 0.905 | 0.944 | 0.746 | 0.846 | 0.912 | 0.600 |
| | LVAD | ECCV-22 | 0.948 | 0.970 | 0.842 | 0.923 | 0.956 | 0.769 | 0.977 | 0.988 | 0.936 | 0.864 | 0.921 | 0.656 | 0.907 | 0.945 | 0.745 | 0.860 | 0.921 | 0.615 |
| | SLAD | ICML-23 | 0.923 | 0.953 | 0.774 | 0.919 | 0.954 | 0.729 | 0.888 | 0.943 | 0.748 | 0.861 | 0.920 | 0.633 | 0.899 | 0.942 | 0.725 | 0.846 | 0.914 | 0.568 |
| | DeepIF | TKDE-23 | 0.858 | 0.921 | 0.647 | 0.866 | 0.928 | 0.639 | 0.874 | 0.940 | 0.709 | 0.801 | 0.890 | 0.544 | 0.808 | 0.899 | 0.556 | 0.765 | 0.875 | 0.462 |
| | DEL | — | 0.980 | 0.992 | 0.899 | 0.981 | 0.992 | 0.900 | 0.983 | 0.994 | 0.946 | 0.942 | 0.973 | 0.787 | 0.947 | 0.976 | 0.820 | 0.928 | 0.961 | 0.771 |

2017). For RGB datasets such as STL-10 (Coates et al., 2011), Internet (Lin et al., 2021), Caltech101 (Fei-Fei et al., 2006), CIFAR-10 (Krizhevsky et al., 2009), CIFAR-100 (Krizhevsky et al., 2009), MIT-Places-Small (Zhou et al., 2017), and MVTec-AD (Bergmann et al., 2019), we utilize two prominent feature extraction backbones: ResNet-50[3] (He et al., 2016) pretrained on ImageNet (He et al., 2019), and CLIP (Radford et al., 2021). In our evaluations, each class within a certain benchmark is alternately treated as inliers, and instances from the other classes are considered outliers. The target dataset comprises all inliers and a portion of randomly selected outliers. Results for each benchmark dataset are averaged across all classes. For every

---

[3]Unless otherwise stated, ResNet-50 is the default feature extractor.

class, we further average the results over a broad range of outlier ratios: $[0.01, 0.1, 0.2, 0.3, 0.4, 0.5]$, which is a more practical setting than some existing works (Wang et al., 2019a; Huyan et al., 2022).

**Competing Outlier Detectors.** We compare the proposed DEL with two themes of outlier detectors: (i) Statistical-based: OC-SVM (Schölkopf et al., 2001), IF (Liu et al., 2008), Shell-Re (Lin et al., 2021), ECOD (Li et al., 2022), LVAD (Lin et al., 2022). (ii) Deep Learning-based: INNE (Bandaragoda et al., 2018), D-SVDD (Ruff et al., 2018), RSRAE (Lai et al., 2019), LUNAR (Lai et al., 2019), GOAD (Bergman & Hoshen, 2020), RCA (Liu et al., 2021), NeuTraL (Qiu et al., 2021), ICL (Shenkar & Wolf, 2021), SLAD (Xu et al., 2023b) and DIF (Xu et al., 2023a). We apply the Ergodic-set normalization (Lin et al., 2022) on these methods if it improves their performance, such as OC-SVM, IF, D-SVDD, and LVAD, etc.

**Competing Thresholding Methods.** We compare with a series of effective threshold learning methods involving Kernel-based: FGD (Qi et al., 2021); Curve-based EB (Friendly et al., 2013); Normality-based: CHAU (Bol'shev & Ubaidullaeva, 1975), DSN (Amagata et al., 2021), OCSVM* (Barbado et al., 2022); Filtering-based: HIST (Thanammal et al., 2014), FILTER (Hashemi et al., 2019); Statistical-based: BOOT (Martin & Roberts, 2006), KARCH (Afsari, 2011), REGR (Aggarwal & Aggarwal, 2017), QMCD (Iouchtchenko et al., 2019), CLF (Barbado et al., 2022) and Transformation-based: MOLL (Keyzer & Sonneveld, 1997) and CLUST (Breunig et al., 2000), CPD (Van den Burg & Williams, 2020).

**Evaluation Metrics.** We evaluate our proposed method on two aspects: ranking accuracy of the generated outlier score function and classification accuracy of the predicted threshold. Specifically, the ranking accuracy is primarily assessed using the Area Under the Receiver Operating Characteristic Curve (AUC). This metric offers a comprehensive evaluation of ranking accuracy, interpreted as the likelihood of an outlier receiving a higher score (Davis & Goadrich, 2006). We also employ AUPR-In and AUPR-Out, which indicate the nuanced ranking accuracy for inliers and outliers, providing a more detailed assessment. Besides, the performance of classification (predicted threshold) is measured with $F_1$-score, a harmonic mean of the precision and recall.

Table 2: Average results on gray-scale datasets.

| Method | MNIST | | | Fashion-MNIST | | |
|---|---|---|---|---|---|---|
| | AUC | PR-I | PR-O | AUC | PR-I | PR-O |
| OCSVM | 0.815 | 0.880 | 0.564 | 0.828 | 0.886 | 0.636 |
| IF | 0.769 | 0.889 | 0.425 | 0.853 | 0.921 | 0.606 |
| INNE | 0.872 | 0.930 | 0.621 | 0.796 | 0.894 | 0.456 |
| RCA | 0.871 | 0.935 | 0.611 | 0.879 | 0.931 | 0.689 |
| D-SVDD | 0.561 | 0.770 | 0.299 | 0.568 | 0.777 | 0.313 |
| RSRAE | 0.871 | 0.917 | 0.667 | 0.781 | 0.889 | 0.523 |
| GOAD | 0.866 | 0.927 | 0.614 | 0.871 | 0.925 | 0.682 |
| Shell-Re | 0.709 | 0.925 | 0.432 | 0.769 | 0.921 | 0.477 |
| NeuTraL | 0.766 | 0.849 | 0.504 | 0.787 | 0.874 | 0.526 |
| ICL | 0.865 | 0.918 | 0.662 | 0.812 | 0.888 | 0.603 |
| LUNAR | 0.761 | 0.815 | 0.581 | 0.719 | 0.827 | 0.451 |
| ECOD | 0.722 | 0.841 | 0.436 | 0.801 | 0.896 | 0.536 |
| LVAD | 0.898 | 0.944 | 0.670 | 0.849 | 0.916 | 0.646 |
| SLAD | 0.845 | 0.909 | 0.592 | 0.839 | 0.909 | 0.604 |
| DeepIF | 0.811 | 0.898 | 0.527 | 0.806 | 0.888 | 0.613 |
| DEL | 0.906 | 0.959 | 0.667 | 0.873 | 0.931 | 0.682 |

Table 3: AUC results on MVTec-AD dataset.

| Type | Class | ResNet-50 | | | CLIP | | |
|---|---|---|---|---|---|---|---|
| | | LVAD | Shell-Re | DEL | LVAD | Shell-Re | DEL |
| Textures | Carpet | 0.872 | 0.895 | 0.983 | 0.877 | 0.964 | 0.983 |
| | Grid | 0.516 | 0.497 | 0.442 | 0.662 | 0.587 | 0.824 |
| | Leather | 0.997 | 0.997 | 1.000 | 1.000 | 0.998 | 1.000 |
| | Tile | 0.977 | 0.984 | 0.987 | 0.940 | 0.925 | 0.990 |
| | Wood | 0.972 | 0.616 | 0.984 | 0.976 | 0.794 | 0.991 |
| Objects | Bottle | 0.941 | 0.909 | 0.972 | 0.816 | 0.801 | 0.992 |
| | Cable | 0.795 | 0.599 | 0.792 | 0.659 | 0.362 | 0.733 |
| | Capsule | 0.679 | 0.583 | 0.628 | 0.546 | 0.647 | 0.656 |
| | Hazelnut | 0.829 | 0.628 | 0.693 | 0.797 | 0.694 | 0.846 |
| | Metal Nut | 0.798 | 0.585 | 0.717 | 0.703 | 0.677 | 0.836 |
| | Pill | 0.687 | 0.613 | 0.688 | 0.681 | 0.646 | 0.676 |
| | Screw | 0.578 | 0.510 | 0.516 | 0.488 | 0.599 | 0.583 |
| | Toothbrush | 0.699 | 0.609 | 0.891 | 0.714 | 0.663 | 0.776 |
| | Transistor | 0.761 | 0.729 | 0.843 | 0.735 | 0.644 | 0.711 |
| | Zipper | 0.889 | 0.859 | 0.955 | 0.748 | 0.840 | 0.909 |
| | Avg. | 0.799 | 0.707 | 0.806 | 0.756 | 0.723 | 0.834 |

## 5.2 Main Results

**Benchmark Datasets.** Tab. 1 shows the substantial improvements in both efficacy and stability achieved by our proposed DEL framework across a diverse range of outlier ratios and various benchmark datasets. In the majority of our experiments, it attains SOTA results. Notably, even with non-aligned, low-resolution datasets such as CIFAR-10 and CIFAR-100, which are known to present challenges (Perera et al., 2019), our method surpasses the current SOTA AUC scores by margins of 6% and 4%, respectively. Furthermore, we observe a significant enhancement in performance with improved feature representation. For example, for the MIT-Places-Small dataset,

Table 4: AUC results on STL-10.

| Method | Outlier Ratio | | | | | |
|---|---|---|---|---|---|---|
| | 0.01 | 0.1 | 0.2 | 0.3 | 0.4 | 0.5 |
| OCSVM | 0.975 | 0.971 | 0.945 | 0.903 | 0.839 | 0.781 |
| RSRAE | 0.978 | 0.978 | 0.951 | 0.934 | 0.902 | 0.871 |
| ICL | 0.981 | 0.968 | 0.949 | 0.924 | 0.895 | 0.856 |
| GOAD | 0.975 | 0.973 | 0.963 | 0.949 | 0.924 | 0.890 |
| LVAD | 0.981 | 0.979 | 0.968 | 0.951 | 0.919 | 0.885 |
| Shell-Re | 0.628 | 0.738 | 0.906 | 0.953 | 0.986 | 0.981 |
| DEL | 0.976 | 0.979 | 0.975 | 0.985 | 0.989 | 0.979 |

the AUC score improves from 0.877 using the ResNet-50 feature to 0.979 with CLIP. Tab. 2 underscores the promising performance observed on the MNIST and Fashion-MNIST datasets. This is particularly noteworthy considering that our framework operates solely on raw pixel features, employing a simpler approach compared to deep learning solutions. Despite this simplicity, it showcases competitive results, effectively matching or even surpassing the efficacy of more complex deep learning models.

Table 5: Timing is measured with ResNet-50 feature.

| Method | Device | # Samples | |
|---|---|---|---|
| | | 500 (sec.) | 10,000 (sec.) |
| OCSVM | CPU | 0.361 | 239.0 |
| IF | CPU | 0.151 | 0.597 |
| INNE | CPU | 1.315 | 54.68 |
| D-SVDD | GPU | 1.294 | 25.94 |
| RSRAE | GPU | 8.054 | 106.7 |
| GOAD | GPU | 10.20 | 270.8 |
| RCA | GPU | 4.715 | 92.08 |
| Shell-Re | CPU | 0.068 | 5.301 |
| NeuTraL | GPU | 8.172 | 162.7 |
| ICL | GPU | 13.31 | 215.6 |
| LUNAR | GPU | 1.760 | 11.42 |
| ECOD | CPU | 1.081 | 29.75 |
| LVAD | CPU | 1.508 | 520.5 |
| SLAD | GPU | 23.86 | 113.3 |
| DeepIF | GPU | 3.376 | 46.45 |
| DEL | CPU | 0.080 | 5.876 |

Table 6: Average $F_1$-score results with SOTA threshold learning methods on DEL's outlier score function.

| Method | STL-10 | Internet | CIFAR-10 | CIFAR-100 |
|---|---|---|---|---|
| BOOT | 0.438 | 0.459 | 0.401 | 0.428 |
| CHAU | 0.482 | 0.478 | 0.428 | 0.432 |
| CLF | 0.638 | 0.624 | 0.479 | 0.514 |
| CPD | 0.613 | 0.618 | 0.560 | 0.602 |
| DECOMP | 0.607 | 0.615 | 0.553 | 0.591 |
| EB | 0.504 | 0.505 | 0.470 | 0.500 |
| HIST | 0.560 | 0.572 | 0.582 | 0.586 |
| FILTER | 0.611 | 0.538 | 0.599 | 0.583 |
| FGD | 0.652 | 0.659 | 0.519 | 0.522 |
| MOLL | 0.423 | 0.419 | 0.410 | 0.434 |
| META | 0.617 | 0.515 | 0.551 | 0.508 |
| KARCH | 0.660 | 0.671 | 0.585 | 0.611 |
| QMCD | 0.136 | 0.129 | 0.181 | 0.166 |
| REGR | 0.583 | 0.628 | 0.601 | 0.624 |
| OCSVM* | 0.549 | 0.467 | 0.421 | 0.478 |
| DEL | 0.687 | 0.683 | 0.645 | 0.653 |

**Defect Detection.** Datasets of defect detection are usually patch-level contamination instead of semantic-level, which is different from outlier detection benchmarks, e.g. STL-10. To maintain the evaluation consistency, we still use the default feature and our framework statistically outperforms the LVAD and Shell-Re, shown in Tab. 3. As MVTec-AD (Bergmann et al., 2019) is previously utilized for semi-supervised/few-shot scenarios (Roth et al., 2022; Huang et al., 2022; Li et al., 2023), the unsupervised method is usually not the best solution, especially for some challenging categories. However, we still obtain quasi-perfection on a series of classes such as "Carpet" (AUC: 0.983), "Leather" (1.000), "Tile" (0.990), "Wood" (0.991) and "bottle" (0.992).

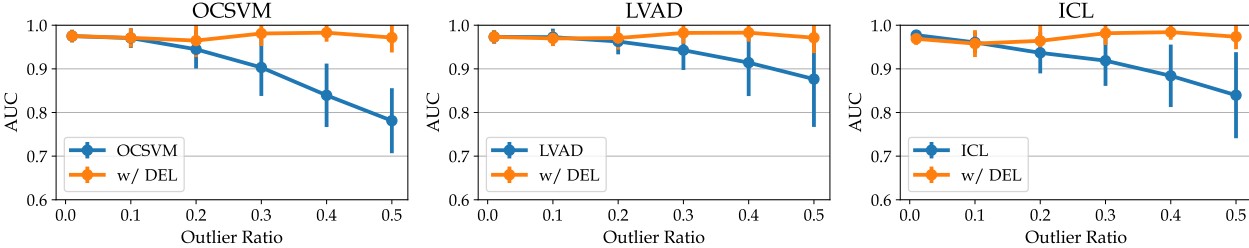

Figure 6: AUC results of integrating DEL with existing methods: OCSVM, LVAD, and ICL. DEL notably enhances the stability of high outlier ratios compared to previously proposed methods.

**Discussion.** To be deemed a desirable unsupervised image outlier detector, the method should demonstrate effectiveness and efficiency across different benchmark datasets (as evaluated in Tab. 1, 2, and 3), various feature representations (as evaluated in Tab. 1 and 2) and a wide range of outlier ratios (as evaluated in Tab. 1, 2 and 4). Notably, many high-performing approaches exhibit domain-specific (e.g., RSRAE, RCA), feature-specific (e.g., IF, ECOD) or outlier ratio-specific (e.g., Shell-Re, LVAD, etc.) characteristics, while DEL appears to strike a good balance across diverse performance scenarios. Despite lacking widely used feature refinement procedures such as self-supervised learning, DEL consistently outperforms existing methods. This validates that default feature representation combined with well-designed distance metrics can yield unexpectedly high accuracy. In Tab. 5, our solution not only excels in accuracy but also speed. Additionally, considering the rapid performance degradation of many existing methods at high outlier ratios,

Fig. 6 demonstrates that our DEL framework enhances stability in such scenarios. Moreover, DEL provides predicted labels, outperforming state-of-the-art threshold learning methods, as illustrated in Tab. 6.

Table 7: Ablation study for different ensemble learning stages of our DEL framework.

| Dataset | Baseline | Baseline+ | DEL |
|---|---|---|---|
| STL-10 | 0.950 | 0.963 | 0.980 |
| CIFAR-10 | 0.870 | 0.919 | 0.942 |
| CIFAR-100 | 0.891 | 0.933 | 0.947 |
| MNIST | 0.861 | 0.877 | 0.906 |

Table 8: Normalization on various distance metrics.

| Dataset | Euclidean Distance | | | Bilateral Distance | | |
|---|---|---|---|---|---|---|
| | w/o N | Z-N | E-N | w/o N | Z-N | E-N |
| STL-10 | 0.884 | 0.884 | 0.950 | 0.955 | 0.962 | 0.964 |
| CIFAR-10 | 0.849 | 0.849 | 0.870 | 0.897 | 0.919 | 0.912 |
| CIFAR-100 | 0.873 | 0.873 | 0.901 | 0.910 | 0.937 | 0.932 |
| MNIST | 0.791 | 0.791 | 0.861 | 0.786 | 0.877 | 0.837 |

### 5.3 Ablation Study

As there remains a concern regarding the extent to which the two ensembling modules contributed to the DEL framework, we conduct ablation studies on four benchmark datasets. AUC results were averaged using two image features: ResNet-50 and CLIP. Compared with the baseline method (Eq. 7), the two ensembling modules improve the performance by $3.2\% - 8.3\%$, shown in Tab. 7. Additionally, we validated the performance of two normalization procedures: Z-score normalization (Eq. 9) and Ergodic-set normalization (Eq. 6), which were respectively paired with bilateral distance and the Euclidean distance with DEL. As depicted in Tab. 8, the choice of normalization procedure is closely linked to the distance metrics used.

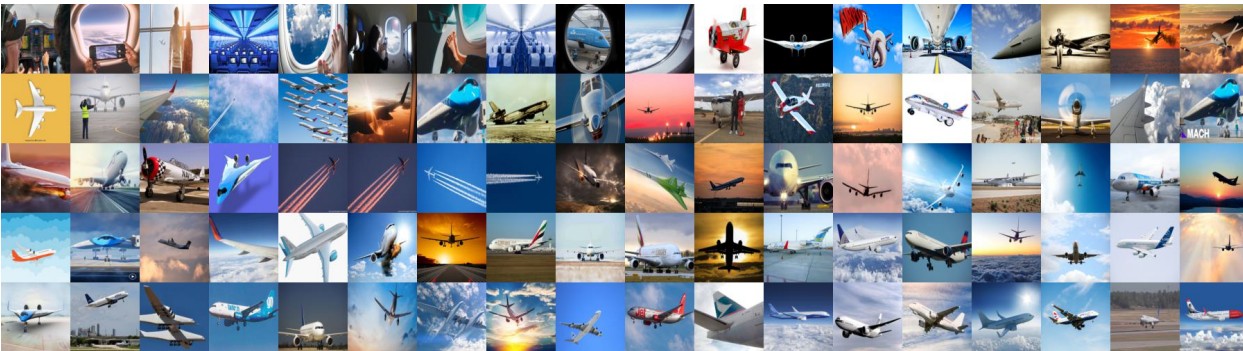

Figure 7: Our ability to identify in-distribution outliers. Images sourced from the Internet dataset. The outlier score decreases progressively from left to right, and top to bottom. Those "strange" airplane images such as passengers, cockpits, and windows demonstrate higher DEL outlier scores than normal airplanes.

### 5.4 Qualitative Result

The outliers identified in our numerical evaluations represent instances that deviate from the inlier distribution. Equally crucial is our method's capability to address in-distribution outliers, such as passengers or cockpit images within the airplane class. Fig. 7 showcases the ranking of internet-crawled airplane images, sorted by their outlier scores. These findings intuitively align with expectations and underscore the potential value of our proposed outlier detector in tasks like data cleaning and organization.

## 6 Conclusion

In this study, our objective is to introduce an effective, stable and efficient unsupervised image outlier detector. To achieve this goal, we propose the DEL, a distance ensemble learning framework. Initially, we establish a simple baseline by comprehensively analyzing the high-dimensional space. Subsequently, we implement a two-stage ensemble learning strategy to enhance stability across diverse target dataset domains and outlier ratios. Overall, DEL demonstrates remarkable accuracy in detection and classification, coupled with efficiency, across a wide spectrum of experimental conditions. Furthermore, DEL stands out for its training-free nature and lightweight design, making it readily deployable in existing methods and real-world visual systems.

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

# A    More Analysis for Distance Ensemble Learning

## A.1    Proof of Lemma 1

**Lemma 1.** Given any $a, b \in \mathbb{R}, |a - b| + |a + b| = 2 \cdot \max\{|a|, |b|\}$.

*Proof. We denote the left side is* $H_1 = |a - b| + |a + b|,$ *and the right side is* $H_2 = 2 \times \max\{|a|, |b|\}$.

$$
\begin{aligned}
When\ a > b > 0, && H_1 = 2a == H_2 = 2a; \\
When\ a > 0 > b\ and\ |a| > |b|, && H_1 = 2a == H_2 = 2a; \\
When\ a > 0 > b\ and\ |a| = |b|, && H_1 = 2a == H_2 = 2a; \\
When\ a > 0 > b\ and\ |a| < |b|, && H_1 = -2b == H_2 = -2b; \\
When\ 0 > a > b, && H_1 = -2b == H_2 = -2b; \\
When\ b > a > 0, && H_1 = 2b == H_2 = 2b; \\
When\ b > 0 > a\ and\ |b| > |a|, && H_1 = 2b == H_2 = 2b; \\
When\ b > 0 > a\ and\ |b| = |a|, && H_1 = 2b == H_2 = 2b; \\
When\ b > 0 > a\ and\ |b| < |a|, && H_1 = -2a == H_2 = -2a; \\
When\ 0 > b > a, && H_1 = -2a == H_2 = -2a.
\end{aligned}
$$

$\square$

## A.2    Parameter Analysis of the Second Ensembling Stage (Ensembling-2)

In Tab. 9, our fixed parameters have overall high performance across different benchmark datasets.

Table 9: Average AUC results are reported with different parameter pairs $\{\phi_1, \phi_2\}$. $\{0.1, 0.3\}$ is the default.

| Dataset | {0.1, 0.3} | {0.1, 0.2} | {0.1, 0.4} | {0.05, 0.1} | {0.05, 0.2} | {0.05, 0.3} | {0.05, 0.4} | {0.2, 0.3} | {0.2, 0.4} |
|---|---|---|---|---|---|---|---|---|---|
| STL-10 | 0.980 | 0.979 | 0.980 | 0.975 | 0.978 | 0.980 | 0.980 | 0.979 | 0.979 |
| CIFAR-10 | 0.942 | 0.939 | 0.946 | 0.934 | 0.939 | 0.943 | 0.945 | 0.941 | 0.944 |
| CIFAR-100 | 0.947 | 0.942 | 0.951 | 0.933 | 0.940 | 0.945 | 0.948 | 0.948 | 0.951 |
| MNIST | 0.906 | 0.899 | 0.899 | 0.887 | 0.895 | 0.898 | 0.896 | 0.900 | 0.898 |

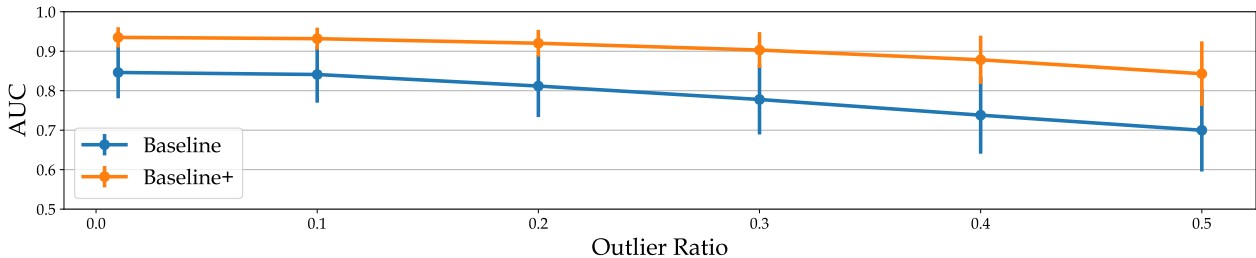

Figure 8: The AUC results comparison between the advance baseline and the baseline on CIFAR-10.

## A.3    Visualization of the First Ensembling Stage (Ensembling-1)

Fig. 8 shows the bilateral distance consistently outperforms the baseline over a wide range of outlier ratios once it is selected. In Fig. 9, 10 11, we show more visualization of the first ensembling stage together with Fig. 3 in the main body. Extensive results validate the efficacy of our distance metrics selection. All results are conducted with the ResNet-50 feature.

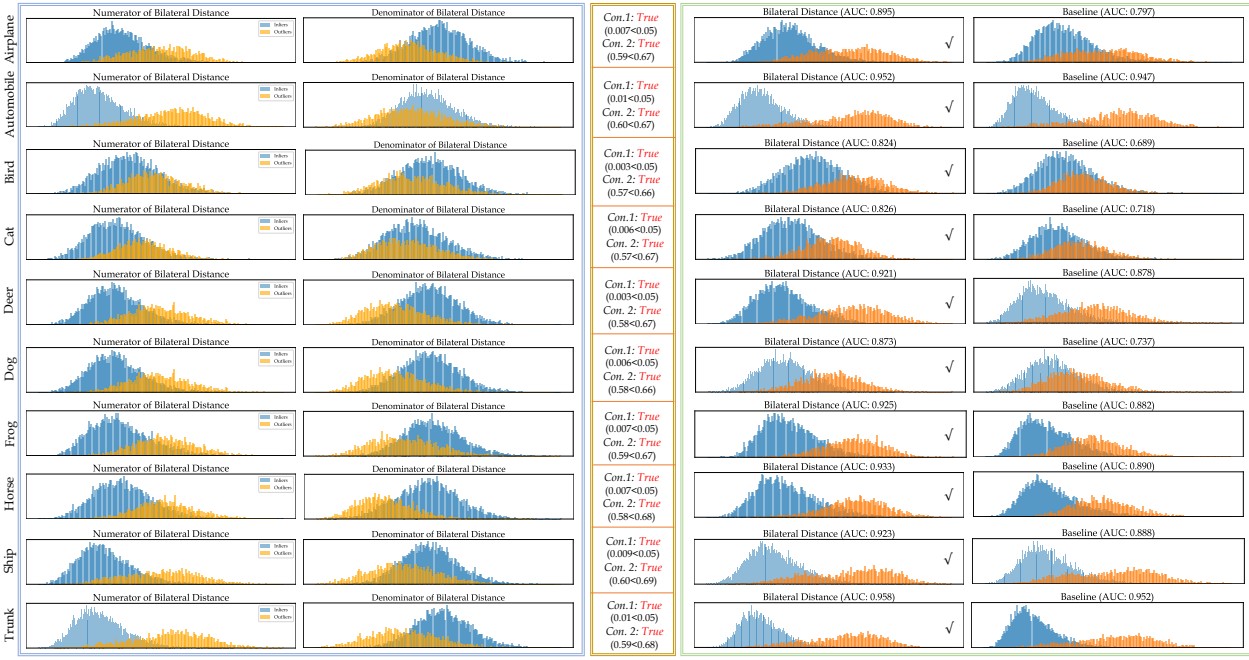

Figure 9: Visualization of the first ensembling stage on CIFAR-10. ✓ refers to the selected distance metrics.

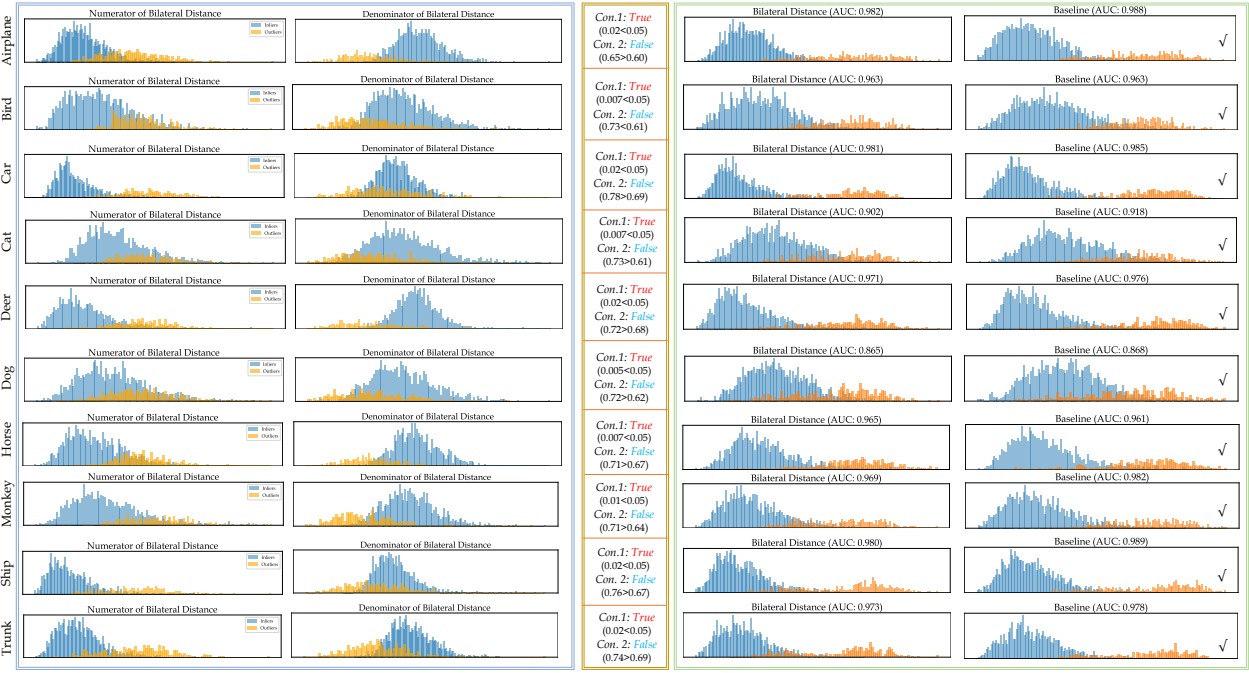

Figure 10: Visualization of the first ensembling stage on STL-10.

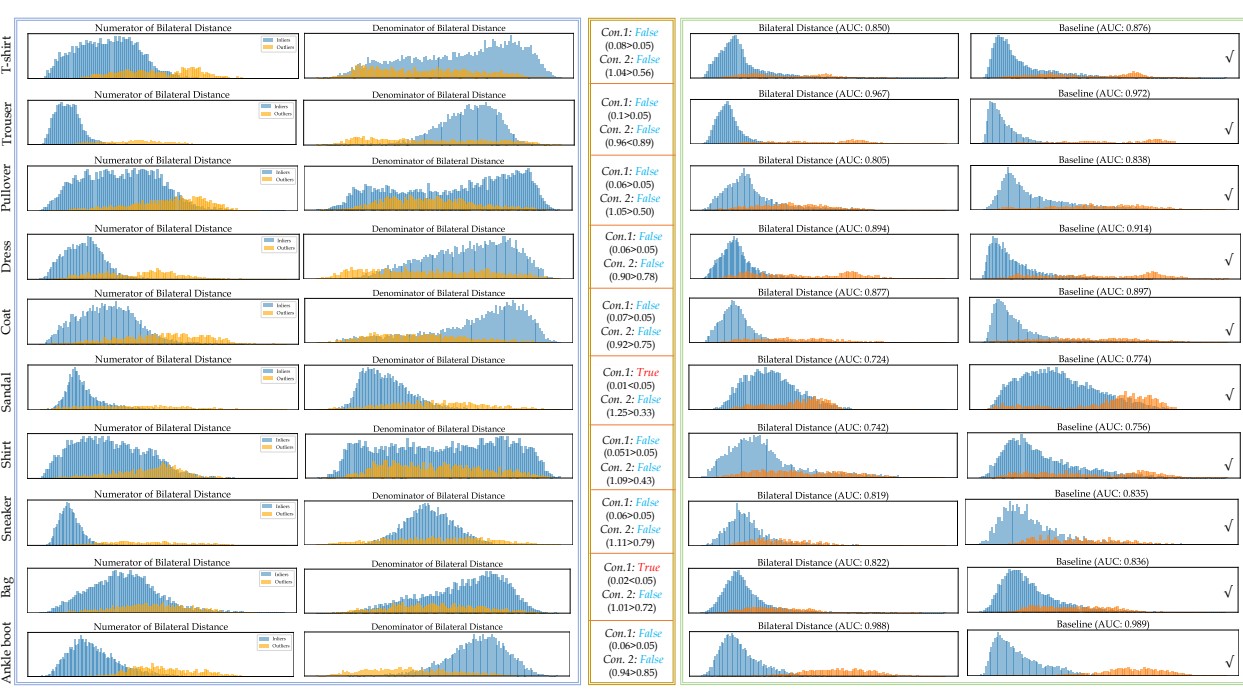

Figure 11: Visualization of the first ensembling stage on Fashion-MNIST.

