# OpenReview forum: "Effective, Stable and Efficient Unsupervised Image Outlier Detection via Distance Ensemble Learning"
_TMLR — Rejected by TMLR_

### Review · Reviewer_sMVR · 2024-05-01

**Summary Of Contributions:**

This paper proposes an anomaly detection approach based on distance. The problem setting defines the anomaly score function for each sample X in a dataset, without assuming another training dataset. The proposed method is a combination of a "baseline" method and an existing method called Shell-Re, both of which are based on the notion of distance from a certain reference point. While the authors stress the importance of ensemble modeling, ensemble models are not clearly defined in Section 4. The definition of the distance-based metrics is based on the distinction between normal samples (inliers) and outliers, which means that the problem setting may not be actually "unsupervised."

**Audience:**

No

**Broader Impact Concerns:**

No.

**Claims And Evidence:**

No

**Requested Changes:**

Due to the lack of a coherent scientific description using the language of statistical machine learning, it is difficult for me to endorse the novelty of this paper. I am uncertain if the paper can be salvaged through partial corrections. Instead, I would recommend completely rewriting the paper, taking into consideration existing works that utilize ensemble and mixture modeling with particular attention to the fact that the notion of distance is automatically taken care of in properly defined probabilistic models. There is no exception to the rule that any academic papers must acknowledge the efforts in the past, even in the pre-deep learning era, and properly position the work in light of prior works.

**Strengths And Weaknesses:**

Strength:
- Provides a simple distance-based anomaly score.

Weakness:
- Incoherent description based on many implicit or ad hoc assumptions. The ratio (8) does not make sense depending on the value of norm(mx), which can be zero vector. No reasonable definition of the distributions introduced is found in p.5, which makes challenging to endorse the value of the proposed framework.
- The novelty is largely questionable. Z-score has been around for a century.
- Ignore most of the existing methods using ensemble and mixture modeling.
- Some of the claims seem to misalign with the generally accepted notion. For example, the use of ensemble modeling has been the standard in practical data analytics since the Netflix competition. In the machine learning community, the curse of dimensionality has been well understood, but it is not considered as something that supports the distance-based notion of an outlier as in Eq. (4).

---

> ### Author Response · Authors · 2024-05-30
>
> Thank you so much for your constructive and insightful comments. We respond to your comments as follows:
>
> - **The problem setting may not be actually "unsupervised."**
> To distinguish whether the problem setting is unsupervised or not depends on whether there is a train-test set split. To propose an unsupervised outlier detector, the understanding or assumption of the natural distinction and definition and outliers between inliers and outliers might be necessary, concerning some related works [1] [2].
>
> - **norm($\mathbf{m}_{\mathbf{X}}$) can be a zero vector.** Although we follow the formula of Z-score normalization, we use an ergodic-set reference vector np.mean(X) rather than the standard mean of the feature matrix np.mean(X, axis = 0). Thus, it will be an origin-closing vector instead of a zero vector.
>
> - **Ignore most of the existing methods using ensemble and mixture modeling.**
> To our best knowledge, the existing ensemble-based outlier detectors involving INNE, IF, DIF (these three methods have been evaluated in our paper), FB (invalid for image data), XGBOD (not unsupervised), LSCP (requires other outlier detectors). We will continually evaluate the latest methods to advance our study.
>
> - **Z-score has been around for a century.**
> Agree. However, Z-score normalization in our method is just an operation, to enhance the discriminativeness of bilateral distance metric.
>
> - **The use of ensemble modeling has been the standard…**
> Although ensemble modeling is not novel in the machine learning community, it is quite challenging but practical in the field of unsupervised image outlier detection. We explain the challenges as follows:
> (1) Unsupervised: no supervision to directly guide us to select the optimal choice; (2) Image-level: some density-based methods are not effective.
>
> **Summary.** We will carefully refine the paper based on your insightful comments. We agree with you that probability/statistical-based ensemble learning methods are commonly used. However, we think ensemble learning is not just about probability/statistics and our proposed DEL belongs to the machine learning ensemble. From Wikipedia‘s definition (https://en.wikipedia.org/wiki/Ensemble_learning): “A machine learning ensemble consists of only a concrete finite set of alternative models, but typically allows for much more flexible structure to exist among those alternatives.” Our method consists of two ensemble learning (auto-selection) stages, to achieve stability across various dataset domains and outlier ratios, respectively. Extensive experimental results illustrate that carefully selecting/ensembling simple distance-based methods can achieve amazing performance, which we believe will be a promising perspective.
>
> ### Reference
>
> [1] Lin, et al. "Locally varying distance transform for unsupervised visual anomaly detection." ECCV, 2022.
>
> [2]  Wang, et al. "E $^{3}$ Outlier: a Self-Supervised Framework for Unsupervised Deep Outlier Detection." TPAMI, 2022.

---

### Review · Reviewer_DQ1Y · 2024-05-11

**Summary Of Contributions:**

The paper studies outlier detection and pruning methods, in the context of image analysis where there is a pre-defined mapping from image to a high-dimensional feature space.
The main insight of the paper is that there are two different regimes for outlier detection in this setting:
 - where the number of outliers is large (e.g., greater than about 20%)
 - where the number of outliers is small

and for each of these two settings, there is different methods that do well empirically.  As such they proposal a two step algorithm:
  1.  estimate the fraction of outliers
  2.  - if the fraction is small, use a form of normalized distance from an all-coordinate mean
       - if the fraction is large, use Re-Shell: (Lin 2021), which iteratively attempts to identify inliers and outliers, and use a normalized distance from outliers.

On several experiments, this approaches shows large improvement over numerous baselines.

**Audience:**

No

**Claims And Evidence:**

No

**Requested Changes:**

For me to feel comfortable accepting this paper, the notation and algorithmic description would need to written in a way that is much more clear.  I am not sure about the theorem, and what it means.  Maybe it can be removed?  See discussion above.

**Strengths And Weaknesses:**

I had a lot of trouble understanding the algorithm due to some confusing points about notation.

 - E-norm(X) seems to take as argument a set $\mathbf{X}$, but its definition treats $\mathbf{X}$ as a vector (e.g., $\mathbf{X} - \mathbf{v}_E$).  What does this mean?
  Then equation (7) applies E-norm on arguments $\mathbf{X}$ a set, and $\mathbf{m_X}$ a vector.

 - Inside E-norm, the values are compared to a vector $\mathbf{v_E}$ where each coordinate is $\frac{1}{n \cdot d} \sum_{i=1}^n \sum_{j=1}^d x_{i,j}$.  How can this possible make sense as a general purpose tool?  For instance, just changing the sign of the third coordinate of each data point does not structurally change the shape or geometry of the point set, but can greatly change this $\mathbf{v}_E$ value.


 - The paper seems to attempt to model data distributions as unimodal, and then prune things that are somehow far from the mean (or a spherical shell around the mean).  In the past decade, there has been very exciting progress in robust mean estimation (which involves these sorts of steps, but with much more rigorous analysis of assumptions and algorithms).  These approaches are very relevant, and seemingly not compared to.  See for instance the book by Diakonikolas and Kane (https://sites.google.com/view/ars-book/)


 - I could not understand what was being proved, or how the argument works in Theorem 1.
   + what does "conditionally symmetric" mean?
   + the proof does not explain **why** or **how** ``Eq 10 can be extended" as claimed in (11)
   + what does "independently symmetric" mean on the last line of the proof?

 - Before equation (13), why do you enforce that "where we constrain its value as less than 0.05"?   I am not sure I understand what this means.

 - the 3-sigma rule makes sense for centrally symmetric data under certain generative assumptions, such as that it is unimodal.  I don't see the assumptions that you are making in this paper clearly stated, so it leaves doubt that this as a useful rule to apply.


By this point, I was fairly lost.



I do think the experimental results seem very good, and indicate that these ideas may be worth publishing.  However, I have trouble following the algorithms, and analysis, and do not feel comfortable with the description of methods in this paper to recommend that it supports its claims, and as a result, may not be of interesting to the community.

---

> ### Author Response · Authors · 2024-05-31
>
> Firstly, we sincerely thank you for your helpful comments and appreciate the recognition of our solid experiments and significant improvements. We then address your concerns about the algorithm as follows:
>
> - **E-norm(X)...**
> The Ergodic-set normalization procedure was originally proposed in LVAD [1], where some details and explanations can be found. In this paper, the Eq (6) is an informal broadcasting formula. By right, the normalization should be performed on each instance instead of the entire target dataset and we will refine it in the revised version.
>
> - **$\mathbf{v}_{\text{E}}$** should be $\[ \mathbf{v}_{\text{E}}[1], \cdots \]$, where each dimension is the same.
>
> - **Robust mean estimation.** Thank you so much for pointing out the robust mean estimation techniques. In this study, our perspective mainly focuses on understanding distance-based methods. By comparing our advanced baseline and Shell-Re (TPAMI-21) [2], we not only can address the mean-shift problem (which might be addressed by robust mean estimator) but also can estimate the approximate outlier ratio that is a critical knowledge in the unsupervised outlier detection.
>
> - **Theorem 1** is about analyzing how the bilateral distance will naturally separate the inliers and outliers distributions.
>
> - "Conditionally symmetric" means whether the two distributions satisfy two conditions: one condition controls the similarity (value as less than 0.05) while another controls the independency. If that is, we select the bilateral distance.
>
> - With respect to Fig. 3,
> if the numerator (left upper plot) and denominator (right upper plot) are independently symmetric, i.e., they do not have distribution overlap and the two distributions are similar, if we take their ratio, i.e, the bilateral distance, the discriminativeness will be notably enhanced (as shown in left bottom plot), compared with the baseline (right bottom plot).
>
> - **Why or how "Eq (10) can be extended" as claimed in Eq (11).**
>
> $\frac{(\left|\mathbf{x}_{i,0}-\mathbf{m}_{\mathbf{X}}[0]\right|)+(\left|\mathbf{x}_{i,0}+\mathbf{m}_{\mathbf{X}}[0]\right|)}{2} =  \max \left \{ |\mathbf{x}_{i,0}|, |\mathbf{m}_{\mathbf{X}}[0]| \right \}$
>
> $\frac{(\left|\mathbf{x}_{i,1}-\mathbf{m}_{\mathbf{X}}[1]\right|)+(\left|\mathbf{x}_{i,1}+\mathbf{m}_{\mathbf{X}}[1]\right|)}{2} =  \max \left \{ |\mathbf{x}_{i,1}|, |\mathbf{m}_{\mathbf{X}}[1]| \right \}$
>
> ...
>
> $\frac{(\left|\mathbf{x}_{i,d}-\mathbf{m}_{\mathbf{X}}[d]\right|)+(\left|\mathbf{x}_{i,d}+\mathbf{m}_{\mathbf{X}}[d]\right|)}{2} =  \max \left \{ |\mathbf{x}_{i,d}|, |\mathbf{m}_{\mathbf{X}}[d]| \right \}$
>
> If we summing up both left and right, we will get:
>
> left = $\frac{\sum_{j=1}^d (\left|\mathbf{x}_{i,j}-\mathbf{m}_{\mathbf{X}}[j]\right|)+\sum_{j=1}^d (\left|\mathbf{x}_{i,j}+\mathbf{m}_{\mathbf{X}}[j]\right|)}{2}$ =
> $\frac{\displaystyle || \mathbf{x}_i -\mathbf{m}_{\mathbf{X}}||_1 + \displaystyle || \mathbf{x}_i +\mathbf{m}_{\mathbf{X}} ||_1}{2}$;
>
> right = $\max \left \{ \sum_{j=1}^d \mathbf{x}_{i, j}, \sum_{j=1}^d \mathbf{m}_{\mathbf{X}}[j]  \right \}$ = $\max \left \{ \displaystyle || \mathbf{x}_{i}||_1, \displaystyle || \mathbf{m}_{\mathbf{X}} ||_1 \right \}$ = $\max \left \{ \displaystyle || \mathbf{x}_{i}-\mathbf{o} ||_1, \displaystyle || \mathbf{m}_{\mathbf{X}}-\mathbf{o} ||_1 \right \}$.
>
> Thus, we will have the Eq (11).
>
> - We take a distance-to-the-mean as the outlier score, which will satisfy a single-Gaussian-like distribution, as illustrated in Shell Theory [2], such that we adopt the "3-sigma" rule.
>
> ### Reference
>
> [1] Lin, et al. "Locally varying distance transform for unsupervised visual anomaly detection." ECCV, 2022.
>
> [2] Lin, et al. "Shell theory: A statistical model of reality." TPAMI, 2021.

---

> > ### Comment · Reviewer_DQ1Y · 2024-06-01
> >
> > I still do not understand a precise definition of "conditionally symmetric."  If this is a formal statement that is to be proved, it needs a very clear definition.
> >
> > I am still confused on getting equation (11), as the RHS should start with the sum outside the max, and then end up with the sum inside the max.
> >   11 = max(1,5) + max(6,2)   != max(1+6,  5+2) = 7

---

> > > ### Author Response · Authors · 2024-06-01
> > >
> > > We appreciate a lot for your reply and your insightful comments do help and guide us to polish this work.
> > >
> > > **conditionally symmetric**: First, we attempt to explain this again. We are going to prove that: if some conditions ($\displaystyle || \text{Z-norm}(\mathbf{X})-\mathbf{o}||_1$ is a constant and $\displaystyle || \text{Z-norm}(\mathbf{X})-\mathbf{o}||_1 \geq \max(\displaystyle || \text{Z-norm}(\mathbf{X}) -\text{Z-norm}(\mathbf{m}_{\mathbf{X}})||_1)$) are satisfied, N1 (the numerator) and N2 (the denominator) will be symmetric about each other. The terminology "conditionally symmetric" originally comes from chaos systems: it describes a special symmetry, where the polarity balance is reconstructed depending on variable reversal and inverse function from offset boosting [1], and we use it in this paper to help readers better understand the property of the bilateral distance. However, we agree with the reviewer that it might cause some complexity to clarify our methodology that can be put into the appendix or removed in the revised version.
> > >
> > > **Eq (11):** Please note that the operation ||.||_1 in Eq (11) is L1-norm (Manhattan Distance, i.e., the sum of absolute differences between points across all the dimensions.) instead of the abs operation |.|. For example: A = [1, 3], B = [0, 1], left = (||A - B||_1+||A + B||_1)/2 = ((|1-0|+|3-1|) + (|1+0|+|3+1|))/2 = 4; left = max{(|1-0|+|3-0|), (|0-0|+|1-0|)} = max{4, 1} = 4, so left==right.
> > >
> > > ### Reference
> > >
> > > [1] Li, et al. "Constructing conditional symmetry in symmetric chaotic systems." Chaos, Solitons & Fractals, 2022.

---

### Review · Reviewer_gexa · 2024-05-16

**Summary Of Contributions:**

This paper proposes a Distance Ensemble Learning (DEL) framework to address instability and complexity in unsupervised image outlier detection. The DEL framework uses a normalized Euclidean distance relative to the dataset mean as a baseline, enhanced by a conditional bilateral distance metric for stability and a high-ratio specific distance transformer called Shell-Re to address mean-shift problems at high outlier ratios. The framework achieves state-of-the-art results on various benchmarks and is significantly faster in inference.

**Audience:**

Yes

**Claims And Evidence:**

Yes

**Requested Changes:**

Please refer to the above comments.

**Strengths And Weaknesses:**

Paper Strengths:

1.	DEL is described as training-free, efficient, and significantly faster in inference compared to existing methods, which is a notable advantage in practical applications.

2.	The framework is rigorously tested across nine benchmark datasets and three feature representations, covering six outlier ratios. This thorough evaluation adds robustness to the findings.




Paper Weakness:

1.	Although the paper claims its efficiency in table 5, a more detailed analysis of the computational complexity, including time and space requirements, would be valuable.

2.	The paper focuses on image data. Exploring the generalizability of the DEL framework to other types of data, such as text or time series, would enhance its applicability.

3.	The task of unsupervised image outlier detection is similar to unsupervised domain adaptation. The authors may find inspiration in the approaches discussed in [1].

[1] Shi, Kuo, et al. "Unsupervised Domain Adaptation Enhanced by Fuzzy Prompt Learning." IEEE Transactions on Fuzzy Systems (2024).

---

> ### Author Response · Authors · 2024-05-31
>
> Thank you for constructive and insightful comments. We response to your questions and suggestions as follows:
>
> - **Detailed analysis of computational complexity.**
> Since this study is non-deep learning based, so we will not make such FLOPS comparison. We have conducted a comparison of CPU memory costs using our method against LVAD [1] in the below table, specifically on the CIFAR-10 dataset employing ResNet-50 features.
> | Method     | LVAD |    Ours |
> | :---        |    :----:   |       :----:   |
> | Memory (mb)      |  0.254      |     **0.204**    |
>
> - **Implementation on other types of data.**
> We have conducted more evaluation on several tabular datasets (sourced from AD-Bench [2]) in the below table:
> | Method      | breastw | pendigits | shuttle |  WBC| WDBC|
> | :---         |    :----:   |            ---: |   ---: |  ---: |   ---: |
> | Ours      |     **0.988**                |    **0.919**     |     **0.991**    |    **0.977**     |    **0.978**      |
> | LVAD     |        0.963                      |       0.882                 |       0.879       |          0.837     |      0.954         |
>
> Compared with the recent baseline model LVAD, our method still demonstrates efficacy on other types of data.
> - **Relation with unsupervised domain adaptation.**
> Thanks a lot for notifying us that, we do learn some important insights from the related unsupervised domain adaption field, and we will cite and discuss it in our revised paper.
>
> ### Reference
>
> [1] Lin, et al. "Locally varying distance transform for unsupervised visual anomaly detection." ECCV, 2022.
>
> [2] Han, et al. "Adbench: Anomaly detection benchmark."  NIPS, 2022.

---

### Decision · Action_Editor_o6gW · 2024-07-03

**Recommendation:** Reject

**Comment:**

Two of the reviewers found that the claims on performance and efficiency could be supported by more evidence (see Claims and Evidence above and comments from reviewers for more details).

Additionally, two of the reviewers have issues with the clarity of the paper (more in comments from reviewers).   One reviewer generally has trouble understanding the paper, particularly, he would like a precise definition of "conditionally symmetric" and clarification on RHS of Eq 1.  Another reviewer would like a reasonable definition of the distributions introduced in p.5.

Two reviewers recommend "reject" and one recommends "leaning to accept".  Overall, I recommend rejection.

**Audience:**

Due to the weaknesses in Claims and Evidence, the audience might not find the paper interesting.

**Claims And Evidence:**

Some of the reviewers are concerned that some of the claims on performance and efficiency could be supported with more evidence.   For example, on performance, one reviewer would like a comparison with existing ensemble and mixture modeling methods in the statistical learning literature and another reviewer would like a comparison with unsupervised domain adaptation, which is similar to unsupervised outlier detection.  On efficiency, one reviewer would like to see a computational complexity analysis.

**Resubmission Of Major Revision:**

The authors may consider submitting a major revision at a later time.